# Monitoring Accumulated Training and Match Load in Football: A Systematic Review

**DOI:** 10.3390/ijerph18083906

**Published:** 2021-04-08

**Authors:** José E. Teixeira, Pedro Forte, Ricardo Ferraz, Miguel Leal, Joana Ribeiro, António J. Silva, Tiago M. Barbosa, António M. Monteiro

**Affiliations:** 1Research Centre in Sports Sciences, Health and Human Development, 5001-801 Vila Real, Portugal; pedromiguel.forte@iscedouro.pt (P.F.); rmpf@ubi.pt (R.F.); ajsilva@utad.pt (A.J.S.); barbosa@ipb.pt (T.M.B.); mmonteiro@ipb.pt (A.M.M.); 2Department of Sports, Exercise and Health Sciences, University of Trás-os-Montes e Alto Douro, 5001-801 Vila Real, Portugal; 3Departamento de Ciências do Desporto e Educação Física, Instituto Politécnico de Bragança, 5300-253 Bragança, Portugal; 4Department of Sports, Douro Higher Institute of Educational Sciences, 4560-708 Penafiel, Portugal; amnfla@gmail.com (M.L.); joana.ribeiro@iscedouro.pt (J.R.); 5Department of Sports Sciences, University of Beira Interior, 6201-001 Covilhã, Portugal

**Keywords:** performance, periodization, training control, match demands

## Abstract

(1) Background: Training load monitoring has become a relevant research-practice gap to control training and match demands in team sports. However, there are no systematic reviews about accumulated training and match load in football. (2) Methods: Following the preferred reporting item for systematic reviews and meta-analyses (PRISMA), a systematic search of relevant English-language articles was performed from earliest record to March 2020. The search included descriptors relevant to football, training load, and periodization. (3) Results: The literature search returned 7972 articles (WoS = 1204; Pub-Med = 869, SCOPUS = 5083, and SportDiscus = 816). After screening, 36 full-text articles met the inclusion criteria and were reviewed. Eleven of the included articles analyzed weekly training load distribution; fourteen, the weekly training load and match load distribution; and eleven were about internal and external load relationships during training. The reviewed articles were based on short-telemetry systems (*n* = 12), global positioning tracking systems (*n* = 25), local position measurement systems (*n* = 3), and multiple-camera systems (*n* = 3). External load measures were quantified with distance and covered distance in different speed zones (*n* = 27), acceleration and deceleration (*n* = 13) thresholds, accelerometer metrics (*n* = 11), metabolic power output (*n* = 4), and ratios/scores (*n* = 6). Additionally, the internal load measures were reported with perceived exertion (*n* = 16); heart-rate-based measures were reported in twelve studies (*n* = 12). (4) Conclusions: The weekly microcycle presented a high loading variation and a limited variation across a competitive season. The magnitude of loading variation seems to be influenced by the type of week, player’s starting status, playing positions, age group, training mode and contextual variables. The literature has focused mainly on professional men; future research should be on the youth and female accumulated training/match load monitoring.

## 1. Introduction

Football is a team sport characterized by intermittent efforts, combining high-speeds and intensity with low-intensity periods [1,2]. Knowing about the match physical and physiological demands allows to carry out the training mode [3]. The training process requires a systematic and periodized application to ensure optimal adaptations to physiological responses and biochemical stresses [4,5]. Researchers and practitioners aim to promote favorable performance outcomes and an adequate recovery for match demands [5]. Football training programs may improve aerobic and anaerobic fitness; these adaptations should be monitored and controlled periodically [6]. The training load has been defined as an input variable for training outcomes, allowing to control training session demands in real time and after each training sessions [7]. The training load can be split up into external (physical) and internal (physiological) load, providing insights about dose-response [6,7]. The external load is defined as the performed work during training sessions or competition, regardless of the internal characteristics. The external load can be monitored by global positioning systems (GPS) tracking systems [8], micro-electromechanical systems (MEMS) [9], local position measurement (LPM), and computerized-video systems [10]. Commonly, external load measures are power output, distances, speeds, accelerations/decelerations, time-motion analysis, and neuromuscular function [5,11]. The internal load refers to physiological and psychological stress and is possible to assess by objective and subjective instruments [5,7]. The most commonly used objective measures are the physiological, such as heart rate, lactate, or oxygen consumption; and training impulse (TRIMP). Moreover, subjective measures usually include ratings of perceived exertion, wellness questionnaires, and psychological inventories [5,12].

The training effects depend on physiological stimulus by intensity, duration, frequency, and recovery periods [6,13]. The external load provides training quality, quantity, and organization; quantifying their components allows an overview of training prescription [4,14]. The physiological adaptations have been well documented [1,2]. However, there is no unique physiological marker that can be used to assess the fitness-fatigue binomial to predict performance [12]. Combining internal and external load data can be used as an approach to overcome the conceptual barrier concerning the fitness-fatigue binomial [15]. However, there is no consensus of an effectiveness monitoring system in professional football [16]. The training load quantification in team sports is often mentioned as a great challenge. This may be due to the difficulty of accurately assessing the skilled performance and cognitive load that influences decision-making [17]. Furthermore, the diversity of monitoring tools appears to have created confusion in dose-response considerations. Indeed, turning these data into relevant information has become a significant challenge to coaches and sport scientists [18].

Currently, a growing number of articles have been published on training load. Recent reviews and meta-analysis focused on team sports aimed to evaluate the association between loading and performance [19,20], intensity [21], training outcomes [22], acute/residual fatigue [23,24], and injury, illness, and soreness [24]. The use of micro-technology to collect and interpret training load has been largely revised in team sports [25,26] and particularly in professional football [27]. Youth football has also been revised with the objective of match running performance [28] and injury incidence [29]. The match running performance has been widely described considering playing position, formation, and opposition standard [30,31,32,33]. However, there have been no previously published systematic reviews and/or meta-analyses about monitoring accumulated training and match load [24]. The match-play represents the greatest physiological stimulus and represents the primary performance outcome [32]. Nonetheless, nearly 80% of the weekly training load results from the training sessions whereas about 20% is from the match-play [1,34]. Understanding the cumulative effect of training is essential to guide the individual athlete’s performance [3,5].

Cumulative effect is a primary factor for the long-term training process and athletic preparation [35]. Training load monitoring plays an important role in training periodization and evaluating cumulative effects variation is essential to an effective training planning according to the individualization principle [36]. Previous research has focused on match load [37] or quantifying training load in specific training moments and highly controlled situations using constrained tasks [38,39]. Monitoring gross and temporal demands during training sessions may help to improve ecological validity. Even more, it may allow to supply an accurate understanding about the inclusion of training load measures in training practices and match management [15,35]. However, there is a lack of consensus on the most effective strategies and training load metrics to measure accumulative training and match demands [16]. Additionally, the different methodologies could lead to outcome differences and bias in the loading analysis [40]. Understanding the seasonal training/match load variations and the relationships between measures would appear important to define the most appropriate monitoring strategy. Therefore, the purpose of this systematic review was three-fold; (1) to analyze intra and inter-individual accumulative training load distribution within a week (microcycle), weeks (mesocycle), and/or season phases; (2) to analyze the intra and inter-individual accumulative training and match load distribution within a week (microcycle), weeks (mesocycle), and/or season phases; and (3) to analyze relationships between internal and external load measures in the accumulative training load quantification.

## 2. Materials and Methods

The present systematic review protocol was registered at the International Platform of Registered Systematic Review and Meta-Analysis Protocols with the number 202080095 and doi:10.37766/inplasy2020.8.0095.

### 2.1. Literature Search Strategy

The preferred reporting items for systematic reviews and meta-analyses (PRISMA) guidelines and the population-intervention-comparators-outcomes (PICOS) design were followed to conduct this systematic review [41,42]. The literature search was based on four databases: PubMed/Medline, Web of Science (WoS, including all Web of Science Core Collection: Citation Indexes), SCOPUS, and SportsDiscus. The eligibility criteria were assured by a PICOS approach and the following search strategy was defined: (1) population: adult and youth football players (participants aged < 13 years); (2) intervention: quantify and compare external (physical) and internal (physiological) load during at least a 1-week period (microcycle); (3) comparison: periodization structure (microcycle, mesocycle, and/or season phase); (4) outcomes: intra- and inter-individual accumulative load distribution; and (5) study design: experimental and quasi-experimental trials (e.g., randomized controlled trial, cohort studies, or cross-sectional studies).

According to the search strategy, studies from January 1980 to March 2020 were included for relevant publications using keywords presented in Table 1. In addition, the keywords were searched with a Boolean phrase (Table 1).

The literature search was accessed during February and March 2020. The search strategy was independently conducted by one review author and checked by a second author. Discrepancies between the authors in the study selection were solved with support of a third reviewer. The authors did not prioritize authors or journals.

### 2.2. Selection Criteria

The included studies in the present review followed these inclusion criteria: (1) training load monitoring studies with adult and youth football players of both sexes; (2) studies with screening procedures based on internal and/or external load measures; (3) only studies that included the training load quantification of gross and temporal demands in complete/full training sessions (with or without match-play load); (4) observational prospective cohort, case-control, and/or cross sectorial design study including at least one week of monitoring; (5) studies of human physical and physiological performance in Sport Science and as scope; (6) original article published in a peer-review journal; (7) full text available in English; and (8) article reported sample and screening procedures (e.g., data collection, study design, instruments, and the outcomes).

The exclusion criteria were: (1) training load-based studies from team sport or football code population (e.g., Australian Football, Gaelic Football, Union, and/or Seven Rugby); (2) studies that monitored only match-play load; (3) participants aged < 13 years and a match format other than 11-a-side football; (4) studies with screening procedures focused on biochemical loading, well-being, and/or injury intervention protocols; (5) studies that included the training load quantification based on field based test and laboratory test; (6) studies that included less than a week of monitoring and experimental trials or study cohort intervention with control group (pre- and post-) to evaluate the effect of a specific training method/program (e.g., small sided games, high intensity interval training, simulated games, or individualized approach); (7) others research areas and non-human participants; (8) articles with bad quality in the description of study sample and screening procedures (e.g., data collection, study design, instruments, and the measures) according to the strengthening the reporting of observational studies in epidemiology (STROBE) statement; and (9) reviews, abstract/papers conference, surveys, opinion pieces, commentaries, books, periodicals, editorials, case studies, non-peer-reviewed text, or Master’s and/or doctoral thesis.

The search was limited to original articles published online until December 2020. Duplicated articles were identified and eliminated prior to application of the selection criteria (inclusion and exclusion). Titles and abstracts were initially selected and excluded according to selection criteria. The selection of full texts was based on a selection to determine the final status: inclusion or exclusion. Disagreements were resolved through discussion between two authors, or via a third researcher if required. Secondary-sourced articles considered relevant and with the same screening procedures were added.

### 2.3. Quality Assessment

The methodological quality was assessed using STROBE statement by two authors [43,44]. This checklist was used in previous reviews due their accuracy in the reporting of observational studies’ cohorts, case-control, and cross-sectional studies [45,46]. The studies were classified as high-quality when missing fewer than three criteria of the STROBE checklist, while low-quality studies were defined as studies missing three or more criteria [45]. It included 22 items: title of the article and abstract interlinked (item 1), introduction (items 2 and 3), methods (items 4 to 12), results (items 13 to 17), discussion (items 18 to 21), and any other information (item 22). Four items were specific to the study design: participants (item 6), variables (item 12), descriptive data (item 14), and outcome data (item 15). The quality assessment was based on the attribution of one point for each checklist item if the criteria were evaluated as being complete (1 point), partial (0.5 points), or incomplete (0 points). The sum of the total points counted was divided by the maximum possible (22 items). Each author performed the classification independently with subsequent inter-observer reliability analysis:Kappa index (0.93; 90%) and confidence interval (CI): 0.92–0.95).

### 2.4. Study Coding and Data Extraction

The data extractions from the included articles were performed according to: (1) summary measures describing construct, measure, measurement, thresholds, and/or metric formula with included article reference and further reading (Table 2); (2) subject and study characteristics according publication date, study design, completive level and standard, sample (N), and sex and anthropometric characteristics (stature and body mass) (Table 3); (3) methodological approaches: observations sample (monitoring period, training sessions recorded, trainings/week, training mode, and number of match-play), training load measures/metrics (internal and external load), and device specification (manufacturer model) (Table 4); (4) main findings: study purpose, periodization design, independent variables, findings, practical applications, and future directions. Data reporting were extracted according study purpose, periodization structure, independent variable, findings, and practical applications.

The outcome measures and the statistical procedures used in the included references were inconsistent between studies, making it impossible to group data and perform the meta-analysis. Characterization of participants is reported as mean ± standard deviation, CI, and effect size (ES) wherever possible. In order to clarify the variety of internal and external load measures used in the included studies, Table 2 consolidates the thresholds used by the authors to calculate metric formulas. In addition, the references correspond to the article reviewed and their construct, measure, and methods. The further reading includes the original references used by the reviewed studies to ensure the methodological procedures.

## 3. Results

### 3.1. Search Results and Study Selection

A total of 7972 titles were collected through four database searches (WoS = 1204; Pub-Med = 869; SCOPUS = 5083; and SportDiscus = 816). No articles were identified from additional sources as a potentially relevant and unidentified research strategy. A total of 188 duplicate records were removed, and 884 articles were removed based on the title and abstract according to inclusion and exclusion criteria. A total of 146 full-text articles were assessed for eligibility and 116 were removed. The reasons for exclusion were: (1) studies not related systematic review purpose (*n* = 53); (2) studies not related to football player’s topic (*n* = 13); (3) studies related only to match load/demands (*n* = 7); (4) studies with screening procedures based on biochemical loading, well-being, and/or injury intervention protocols (*n* = 15); (5) studies that included field-based test and laboratory test for training load quantification (*n* = 11); (6) editorials, commentaries, and literature reviews (*n* = 12); (7) case studies (*n* = 3); (8) conference abstract/papers (*n* = 1); and (9) other language (*n* = 1). After screening procedures, 36 articles were included in the present systematic review. A detailed representation of the screening procedures is depicted with a PRISMA flow diagram in Figure 1.

### 3.2. Participant Characteristics

The reviewed articles were published between 2004–2020. All included articles presented a quasi-experimental approach based on observational and prospective cohort design. The included studies were performed in elite/professional (*n* = 32), pre-elite (*n* = 3), and amateur (*n* = 1) football. One article did not specify the participants’ competitive level. Twenty-seven articles focused on adult player population and nine on youths. The geographic location of the populations studied in reviewed studies were Australia (*n* = 1), Brazil (*n* = 1), France (*n* = 3), Italy (*n* = 1), Korea (*n* = 1), Norway (*n* = 2), Portugal (*n* = 6), Spain (*n* = 5), Swiss (*n* = 1), The Netherlands (*n* = 3), and the United Kingdom (*n* = 11). Four studies did not specify the geographic location and one study was sampled in an European population.

The study samples ranged between 13–160 participants. All articles were performed on male football players, except one on female players. A total of 1317 (1302 men and 15 women) adult and youth football players were analyzed for this systematic review. The mean and standard deviation for age and anthropometric data (weight and height) in the included studies was 22.71 ± 4.37 years, 74.13 ± 6.77 kg, and 1.71 ± 0.05 m, respectively. Table 3 provides a summary of the participants demographics.

### 3.3. Quality Assessment

In the evaluation of methodological quality, the mean quality score and standard deviation of all the included studies was 0.79 ± 0.06 (Table 3). One study (3.33%) was classified with a quality score of 0.65. Twenty studies (56.67%) were classified between 0.7 and 0.8, whereas fifteen studies (40.00%) had a quality score between 0.8 and 0.9. None of the reviewed studies had the maximum score (1.0) or below 0.5 (min: 0.65; max: 0.89).

### 3.4. Data Organization

The results are presented in the following three sub topics: (1) analysis of the intra- and inter-individual accumulative training load distribution within one week (microcycle), weeks (mesocycle), and/or season phases; (2) analysis of the intra- and inter-individual accumulative training load and match load distribution within one week (microcycle), weeks (mesocycle), and/or season phases; and (3) analysis of the relationships between internal and external load measures in the accumulative training load quantification. Observation samples were collected from 17 to 2981 training sessions and varied between 3 and 6 trainings per week. Twenty studies analyzed training data and ten articles integrate training data with match load. The monitoring period in the included studies ranged from 3 to 43 weeks. The included match-play varied from 1 to 623 games. Four studies did not describe the number of observed weeks and six studies did not describe training sessions. Eleven articles evaluated training load with internal load measures; fourteen articles included only external load measures; and eleven studies analyzed internal and external measures.

The training load quantification in the included studies were based on internal and external load measures/metrics. Twelve articles analyzed only internal load measures, twelve articles evaluated the external load, and twelve studies assessed both measures. The studies that quantified only internal load were based on summated zones of maximum heart rate (HR_max_) (*n* = 10), and training impulse (*n* = 11). Banister TRIMP was reported in four studies, Edwards TRIMP in five studies, and lactate threshold (LTzone) and modified Stagno training impulse (TRIMP_MOD_) were both required in one study. Still, external load measures were quantified with distance and covered distance in different speed zones (*n* = 27), acceleration and deceleration (ACC/DEC) (*n* = 13), accelerometer metrics (*n* = 11), metabolic power output (*n* = 4), and ratios/scores (*n* = 6).

The methodological approaches of the reviewed articles were based on short-telemetry systems (*n* = 12), GPS systems (*n* = 25), MEMS (*n* = 18), LPM systems (*n* = 3), and multiple-camera systems (i.e., Prozone^®^, Leeds, UK) (*n* = 3). Additionally, the internal load measures were reported with perceived exertion scales (i.e., Borg’s Category-Ratio scale, Hooper Index, and Fatigue Questionnaire) (*n* = 16). The internal load based on heart rate (HR) measures were reported in twelve of the included studies (*n* = 12); with 1 Hz telemetry system and five studies with 5 Hz. Two studies did not specify the telemetry range in the methodology description. Furthermore, internal load based on perceived exertion by Borg’s Category-Ratio scale was presented in fifteen studies. One study assessed perceived exertion with the Hooper Index and one other with the Fatigue Questionnaire. Regarding systems tracking, 5 Hz GPS, 10 Hz GPS, and 15 Hz GPS were used in one study, fifteen studies, and four studies, respectively. The 100 Hz MEMS integrated the GPS device and was reported in ten studies. The LPM system was reported only in one study.

The data organization respected the three main purposes of this systematic review. Table 4 presents the methodological approaches selected by the studies included in this review.

### 3.5. Weekly Training Load Distribution Analysis

Eleven reviewed articles analyzed weekly training load distribution. Two articles included only internal load measures, two articles evaluate only external load, and six studies analyzed both training load measures. Regarding periodization structure, seven studies analyzed weekly microcycle (1-game week), four studies quantified training load over mesocycles (week-block), and three articles included the training load quantification across different seasonal phases. One article did not specify the periodization structure for its analysis. Observations samples were collected from 27 to 2591 training sessions and varied from 4 to 6 training sessions per week. The monitoring period in the included studies ranged between 7 and 42 weeks. The included match-play ranged from 1 to 612 games.

The independent variables in the weekly training distribution analysis were age (*n* = 3), training day (*n* = 7), mesocycle structure (*n* = 3), training mode/type or sub-components (*n* = 1), playing position (*n* = 5), and contextual variables (*n* = 2). Table 5 provides the studies predominantly with a focus on weekly training load distribution analysis.

### 3.6. Weekly Training Load and Match Load Distribution Analysis

Fourteen articles analyzed the weekly training load distribution. Six articles assessed external load, two articles analyzed internal load measures, and two studies assessed both training load measures. Regarding the periodization structure, five studies evaluated the weekly microcycle (1-game week), two studies analyzed three different weekly microcycles (1-, 2- and 3-game week), and three studies quantified training load by mesocycles (week-block). Any article included in this systematic review analyzed weekly training load and match load distribution across different seasonal phases. Observation samples were collected from 10 to 2981 training sessions and varied from 4 to 7 training sessions per week. The monitoring period in the included studies ranged from 3 and 55 weeks. The included match-play varied between 3 to 55 games.

The independent variables applied in weekly training load and match load distribution analysis were age of players (*n* = 1), training day (*n* = 2), weekly microcycle type (*n* = 3), mesocycle structure (*n* = 3), player’s starting status (starters or non-starters) (*n* = 3), training mode/type or sub-components (*n* = 1), and playing position (*n* = 2). Table 6 provides the studies predominantly focusing on weekly training/match load distribution analysis.

### 3.7. Relationships between Weekly Internal and External Load

Eleven articles evaluated internal and external load relationships during training load quantification. Of these, five articles evaluated internal load relationships, five articles compared external load measures, and one study assessed the relationship between internal and external load. Four studies analyzed a weekly microcycle (1-game week) structure, and six articles did not specify the periodization structure. Observation samples were collected from 24 to 1029 training sessions and varied between 2 and 5 training sessions per week. The monitoring period in the included studies went from 9 to 43 weeks. The included match-play varied from 1 and 623 games.

All the eleven included articles in this sub-topic focused on comparison of internal and external load measures during training session. No other has analyzed the internal and external load relationships during match load. The independent variables applied in weekly training load distribution analysis were training day (*n* = 2), mesocycle structure (*n* = 1), training mode/type or sub-components (*n* = 2), playing position (*n* = 2), and training load indicators (*n* = 3). Table 7 provides the studies predominantly focusing on relationships between internal and external load during weekly training load.

## 4. Discussion

The present systematic review focused on three purposes: (1) analyzing intra- and inter-individual accumulative training load distribution within week (microcycle), weeks (mesocycle) and/or season phases; (2) analyzing the intra- and inter-individual accumulative training load and match load distribution within one week (microcycle), weeks (mesocycle), and/or season phases; and (3) analyzing relationships between internal and external load measures in the accumulative training load quantification.

The findings from the reviewed studies were organized into weekly training load distribution analysis, weekly training and match load distribution analysis, and relationships between weekly internal and external distribution. Therefore, the present discussion was conducted following the independent variables of age group, match contextual factors, periodization structures (i.e., microcycles, mesocycles, and/or season phases), playing positions, training mode or sub-components, week schedule format (i.e., 1-, 2- and 3-game week), player’s starting status, playing positions, and training load indicators. This systematic review ensures a general overview about monitoring daily and accumulated load. The main results demonstrated that the weekly microcycle presented a high load variation and a limited variation along season phases. Both were influenced by the type of week, player’s starting status, playing positions, age group, training mode, and contextual factors.

### 4.1. Weekly Training Load Distribution Analysis

The distribution of daily and accumulated load during a weekly microcycle (1-game week) was specified by seven included studies. Of these studies, six studies employed the format «match-day (MD) minus format» (i.e., MD- and/or MD+) and one study subdivided the week into post-match (session after the match), pre-match (session before the match), and mid-week (remaining training sessions). The accumulated training load showed a non-perfect load pattern within weekly microcycle. On that, the literature reported the greatest intensity and volume mid-week. However, there is no consensus among reviewed studies about the training day with highest values for high-intensity movements. On the other hand, a small seasonal load variation was reported with a non-significant higher accumulated weekly physiological load during pre-season. The influence of match-related contextual variables was clearly evidenced, which requires a more individualized approach. The training mode and age-related influence should also be considered for weekly training load distribution.

Clemente et al. [88] noted an intra-week load variance. Clemente et al. [89] reported that the highest load occurred over the MD+2, MD-5, MD-4, and MD-3. The lowest load was found on the MD+1, MD-2, and MD-1. The daily and accumulated load were significantly reduced on the MD-1, with no significant differences observed in other days [56]. Oliveira et al. [72] noted conflicting findings to daily internal and external load. The external loads were similar until MD-1 while the internal load did not present the same pattern. In the same line, MD+1 provided the highest average speed and high-speed running (HSR). Contrarily, MD+1 showed the lowest session rating of perceived exertion (sRPE) score. Malone et al. [56] showed the greatest intensity and covered distances performed on the MD-5 and MD-3. Oliveira et al. [72] presented a non-perfect load pattern by decreasing values until MD-1: MD-5 > MD-4 < MD-3 > MD-2 > MD-1. It was clear that MD-5 and MD-2 provided highest high-intensity [48]. In another study, the highest values were reported on MD-3 relative to the other days (MD-4, MD-2, MD-1) [99]. As well as that, a large weekly variation was found for the same type of day. That may exceed the recommendations to progressively load increase (between 5 and 15%) [91]. Regardless of the stage of development, Coutinho et al. [48] also observed an unloading on the MD-1. Conversely, the weekly training load distribution in the other age groups was different. The U19 showed high values of high-intensity activity in mid-week and pre-match. Moreover, U15 experienced residual weekly training load variations. The weekly external load distribution differs when comparing two teams from different countries [89]. According to Clemente et al. [89], the Portuguese team had a greater training volume on MD-2 and the Dutch team on MD-1. In the same study, significant differences were not found on MD-5 and MD-4 between teams. In the same study, the number of sprints covered during training sessions were different. The Portuguese team completed more sprints on MD-5, MD-3, and MD-2, whereas the Dutch team on MD-5.

The mesocycle or week block was explained in four studies. Brito et al. [69] divided the monitoring of the seasonal training load into four different phases (preparation I, competition I, preparation II, and competition II). Loading variation was reported across the season to sRPE and weekly training load, 5–72% and 4–48%, respectively. The highest sRPE values were observed during match-play, especially the last phase of the season (i.e., competition II). By contrast, fatigue scores did not detect differences along the competitive season. The variation of individual fatigue scores was only reported within the weekly microcycle. As well as that, Oliveira et al. [72] showed similar outcomes with Hooper Index scores across ten mesocycles and within their respective weekly microcycles. In addition, a small seasonal load variation was reported even if there were no significant differences between mesocycles. Clemente et al. [88] reported a small increase in load through descriptive statistics. Owen et al. [99] analyzed seasonal loading using a mesocycle structure (6 × 1-week blocks). No significant variations have been found along mesocycles differing from the weekly microcycle training load variation.

Two studies of this review examined the weekly training load distribution comparing pre-season versus in-season [55,56]. One study focused their analysis in the intra-week variations isolating pre-season and comparing two professional teams. The main findings and conclusions of these three studies were consistent with the studies that opted for the mesocycle structures [56,69,72]. Nonetheless, the study by Malone et al. [56] reported an additional in-season variation to covered distance and higher HR_max_ values during the beginning of the in-season than at the midpoint and endpoint. Jeong et al. [55] noted a higher accumulated weekly physiological load during pre-season when compared to in-season.

The inter-positional variation was examined in five reviewed studies with predominantly weekly training load distribution. Akenhead et al. [53] showed that only total covered distances and ACC/DEC were able to differentiate playing positions. Conversely, HSR and sprinting showed no positional differences. The central midfielders (CM) covered more distance at low and moderate acceleration thresholds than central defenders (CD). Indeed, when expressed in relation to distance covered, the wide defenders (WD) displayed a higher ACC/DEC density than CM. In Malone et al. [56], the CM and WD presented highest and CD the lowest values to total covered distance. Oliveira et al. [72] reported no significant changes for playing positions across the mesocycles analyzed. CM covered highest total distances than other playing positions. However, the authors did not find statistical significance. Moreover, the covered distance at high-intensity threshold proved that the interposition difference only took place in the first microcycle when comparing CD versus WD and WD versus wide midfielder (WM). This suggests that WD and WM have a higher high-intensity training profile. On the other hand, Owen et al. [99] documented significant differences within playing positions, especially before the match-play. CD showed lower covered distance values in comparison to CM and WM. It should be noted that the CM presents the highest covered distance at low intensity. The WD exhibited lower velocities and perceived exertion than CM and WM. The CD covered lower total distances and sprints while the opposite was pointed to WM. The analyses set out by authors revealed a limited positional variation across weekly training load. Oliveira et al. [72] provided a limited positional variation. Indeed, differences were found within-macrocycle whereas the load remained similar at the days of weekly microcycle, with the exception of MD-1.

Two included studies analyzed the time-motion and physiological profile by young football players using training data [47,48]. Coutinho et al. [48] described the age group pattern load over a typical week. Abade et al. [47] presented the overall loading without specifying any periodization structure. There were similar findings in both studies, reporting differences in the physical and physiological demands during training sessions. The under-15 (U15) training sessions had the most regular activity with less physiological demands [47,48]. The under-17 (U17) displayed the highest physical and physiological stimulus and under-19 (U19) had the highest high-intensity activity [48].

The influence of match-related contextual variables was mentioned in two studies [69,108]. Brito et al. [69] noted that the internal load of young football players was affected by contextual factors (i.e., result of previous match, the opponent’s level, and the location of the previous and following marches). According to the authors, the highest accumulated training load occurred during the training sessions after losing or drawing. By contrast, the lower loading was found before and after a match-play with a top-level opponent. After playing an away match-play, weekly training loads were higher than for a home match-play. Nonetheless, Owen et al. [108] did not report significant findings to confirm that contextual factors have an influence although the descriptive data point to a decrease in the training load after a win and away match-play. These findings are consistent with the need for a more individualized approach to initial preparation and subsequent match conditions [119,120]. Previous studies emphasize the importance of considering the independent and interactive effects of match-related contextual factor to the physical component of football performance [121,122].

The influence of the training mode, type, or sub-components were assessed in one study by weekly training load analysis. The training intensity presented associations with technical/tactical specifics and cool-down training sessions during the pre-season [55]. The contextual factors influenced the weekly training load distribution [61]. According to Rago et al. [61], the weekly TL seemed to be slightly affected by match-related contextual variables.

### 4.2. Weekly Training Load and Match Load Distribution Analysis

Two different periodization structures were explained in the studies with predominantly training and match load analysis. The weekly microcycle was reported in six studies and mesocycle (or week-blocks) were used by four authors. In this scope, the studies that included the match load also appear to show differences in the loading distribution, especially in the middle of week (i.e., MD-5, MD-4, and MD3). Limited load variation between the mesocycles were also reported. Furthermore, the type of weekly microcycle (i.e., one-, two-, and three-game week) appears to decidedly influence in the loading distribution. Additionally, the compensatory session was more intense than the recovery session. The match-related contextual factors, playing position, player’s starting status, age-related influence, and training mode should also be considered for weekly training load and match load distribution analysis.

Kelly et al. [70] showed a total distance and sRPE decreased in the MD-3. The high-intensity movements (HSR and sprinting) were higher in MD-3 and MD-2 than MD-1. Another study presented a progressive increase in perceived load until mid-week (i.e., MD-3) and subsequent decrease until MD-1 [68]. Martin-Garcia et al. [82] reported that the overall external load decreased progressively before match-play, especially in the MD-2 and MD-1. In agreement with the based-volume metrics, a reduced load raised in the MD-1 and MD-2 compared to MD-5, MD-3, and MD-4. Owen et al. [99] reported a significantly higher percentage for ACC/DEC values during MD-4, MD-2, and MD-1. Using a multi-modal approach, this study suggests that these metrics may provide higher levels (21% to 48%) when compared to explosive movements (2% to 11%). The compensatory session was more intense than the recovery session [79]. Similarly, Owen et al. [99] demonstrated that the MD-4 and MD-3 were the highest intensity and volume within the weekly microcycle, revealing a weekly highest load closer to MD. Anderson et al. [87] and Sanchez-Sanchez et al. [91] verified the greatest load in MD-3 and MD-2, respectively. The MD-1 was the lowest load in various studies whereas Owen et al. [99] presented a higher ACC/DEC in MD-1 than MD-2. At amateur level, the MD+1 and MD-1 were less loading [91]. The observed main findings seemed to have been converging in a strategy tapering based on a gradual reduction until the last day before MD.

During the competition period, the studies with training and match data seem to indicate a limited load variation between the mesocycles; similar with within playing positions. In contrast, the weekly microcycle presented the reported fluctuations in the external and internal load, which was further influenced by the playing position. Kelly et al. [70] described a slight increase in the beginning of the in-season and a small decrease along season. The total distance and sRPE was greater at the beginning of the in-season. The weekly accumulated load varied during MD-3 and MD-4 (40%), depending on the selected training load measure and playing positions. HSR and sprinting were the metrics that presented the greatest variability within the weekly microcycle (0.80%) [82]. The mid-season also showed a reduction in training volume [70]. On the other hand, the training time and typical weekly training load did not differ within microcycles in amateur football [91]. Wrigley et al. [60] also established that stage of development could influence variations within the weekly microcycle. These findings were similar to those verified in studies without match load [47,48].

The different type of weekly microcycle was analyzed in two reviewed studies [87,95]. Anderson et al. [87] quantified the training and match load during a one-, two-, and three-game week schedule. Clemente et al. [95] describe weekly training load variation in week with five, four, and three training sessions. A daily and accumulated load differed with the type of week schedule [87]. Clemente et al. [95] verified that the typical training intensity in the one- and two-game week schedules were compatible. However, the same did not occur in the three-game week. Therefore, the total accumulative load was lower in the one-game week schedule in comparison with the two- and three-game week schedules. Clemente et al. [95] verified that the accumulated total distances and number of ACC/DEC were three to four times higher than average match demands. The HSR and sprinting were one to two times greater than match demands. This kind of relationship between training and match load (scores/ratios) were studied in two studies included in this systematic review [95,108]. The training/match ratios varied ~2 to 4 arbitrary units (AU) considering external load. These proportions were dependent on the numbers of training sessions per week and that may infer an independence between weekly training load and match demands. The specific multi-modal approach suggested a significant variation in the volume and intensity scores across microcycles [99]. The variability match-to-match was ~16–31% (i.e., HSR and sprinting). Subsequently, it is possible to ensure that these external metrics revealed a greater sensitivity regarding contextual factors and type of week. The microcycle format may improve insights on how to appropriately implement periodization during fixture congestion [18,123]. Indeed, the studies with training and match data also demonstrated a limitation of the accumulated load between playing positions during the season [70,71]. Assessing the load patterns during the weekly microcycle may provide a most accurate positional load comparison [70].

The effect of player’s starting status was explored in five studies of this review. Anderson et al. [106] verified a significant effect of player’s starting playing in the total distance and high-intensity activity. Generally, starters covered more running, HSR, and sprinting distances than fringes and non-starters. Similarly, large to very large differences occurred in the perceived exertion within starters and non-starters [71,97]. The competition time was the main source to these variances [71,106]. The pre-season and winter-break seemed to have the highest variability across playing position [71,105]. Given the consistent these findings, it is reasonable to argue that the starting status could affect physical and physiological profiles [106]. The implementation of complementary training can be a strategy to reduce variance on the non-starting status. Stevens et al. [83] described that training to non-starters was generally higher than regular training sessions. Martin-Garcia et al. [82] verified that the compensatory session may produce the greatest ACC/DEC value within the weekly microcycle. Another interest finding in this study was the marked difference in training load at MD+1 between players completing the majority of the match-load (>60 min) versus players with partial or no playing time (<60 min).

One study aimed to quantify typical weekly training load and their content match load by under-14 (U14), under-16 (16), and under-18 (U18) football players [60]. The results proved that the training intensity and volume increased with age. Additionally, there were significant differences in the weekly loading periodization according the development stage. The weekly field-based load was higher in the U18 than U14 and U16. Moreover, the perceived exertion did not differ within age group. The U14 and U16 training process prioritized technical and physical development, while U18 focused on competition. These conclusions were similar with two other reviewed studies that included only the training data [47,48]. Importantly, the oldest age group in Wrigley et al. [60] adopted an exponential decrease (tapering). Nonetheless, Coutinho et al. [48] visualized this trend only in U17 age group knowing that this study only applies training data. According to these conclusions, it is possible that different stages of development required different load patterns.

The training mode or sub-components were analyzed for only one study, predominantly in training and match load distribution. Their findings showed that weekly field load was higher than total gym-based load [60]. These data may provide valuable information to the strength and conditioning coach about the high intensity active profiles that could be used to develop soccer-specific training drills [40].

### 4.3. Relationships between Weekly Internal and External Load

In this systematic review, the relationships between internal and external load were explained in ten studies. Of these studies, five studies reported the relationship within internal load methods, four studies analyzed internal and external load relationships, and one study compared only external load metrics. The literature evidenced positive within and between-individual correlations for perceived exertion, heart rate-derived measures, and external load indicators for elite female [52], semi-professional [115], elite/professional [50,65], and young amateurs [53,98]. The magnitude of correlations tended to increase when it was considered a within-individual correlation. The sRPE was a consistent method to quantify internal load along an entire season. The internal training load may be useful to assess accumulated training load and the relations with external training load by playing position, training mode, and/or age-groups. The reviewed studies showed a relationship between external and internal training load indicators. However, analyzing high intensity demands must take into account some considerations about speed thresholds, metabolic power output, accelerations, and accelerometers measures.

Alexiou and Coutts [52] reported positive correlation for sRPE with Banister’s TRIMP, LTzone, and Edwards’s TRIMP (r = 0.84, r = 0.83, and r = 0.85, all *p* < 0.01, respectively). Campos-Vazquez et al. [50] also reported correlations between sRPE with TRIMP_MOD_ and Edwards TRIMP (r = 0.92 to 0.98). Casamichana et al. [115] reported associations for Edwards TRIMP with sRPE (r = 0.57 *p* < 0.01, respectively). The correlations presented by Impellizzeri et al. [54] were statistically significant for sRPE and Edwards, Banister, and Lucia TRIMP’s (from r = 0.50 to 0.85, *p* < 0.01). Kelly et al. [65] indicated correlation between changes in sRPE and Edwards TRIMP (r = 0.75, *p* < 0.05). Particularly, these main findings prove that there were correlation changes between perceived exertion and HR measures for elite female [52], semi-professional [115], elite/professional [50,65], and young amateurs [53,98].

Casamichana et al. [115] reported associations for PL with Edwards TRIMP and sRPE methods (r = 0.70 and r = 0.74, all *p* < 0.05, respectively). Total distance covered associated with PL, sRPE, and Edwards TRIMP methods (r = 0.70 and r = 0.74, all *p* < 0.05, respectively). Gaudino et al. [108] reported for RPE with HSR, impacts and accelerations (r = 0.14, r = 0.09 and r = 0.25, all *p* < 0.001, respectively). Additionally, the adjusted correlation for RPE were r = 0.11, r = 0.45, and r = 0.37, respectively. In the study by Rago et al. [107], RPE was moderately correlated to MSR and HSR using the arbitrary method (*p* < 0.05; r = 0.53 to 0.59). However, the magnitude of correlations tended to increase for the individualized method (*p* < 0.05; r = 0.58 to 0.67). When the external load was expressed as percentage of total distance covered, no significant correlations were observed (*p* > 0.05). Scott et al. [57] reported significant correlations for total distance, low-speed running, and PL with the HR-based methods and sRPE (r = 0.71 to 0.84; *p* < 0.01). The internal measures had correlation with volume of HSR and sprinting (r = 0.40 to 0.67; *p* < 0.01). Marynowicz et al. [98] reported a large and positive within-individual correlations for total distance, PL, number of ACC, and sRPE (r = 0.70, 0.64, and 0.62, respectively, *p* < 0.001). Small to moderate within-individual correlations were noted between RPE and measures of intensity (r = 0.16 to 0.39). A moderate within-individual correlation was observed between HSR per minute and RPE (r = 0.39, *p* < 0.001).

Gaudino et al. [90] compared high-intensity activity using total distance covered at speeds > 14.4 km × h^−1^ and the equivalent metabolic power threshold (>20 W × kg^−1^). Measuring high-intensity movements with speed categories may underestimate the energy cost by training sessions and playing positions. Moreover, the difference between methods also decreased as the proportion of high intensity distance within a training session increased (R^2^ = 0.43; *p* < 0.001). Therefore, metabolic power estimations may have higher precision to evaluate physical demands during training sessions.

Vahia et al. [58] was the only study that reported monthly correlations (r = 0.60 to 0.73, (*p* < 0.05) and overall correlation (r = 0.64, *p* < 0.001). The correlations between sRPE and HR-load were found for all months, consequently sRPE is a consistent method to quantify internal load along an entire season.

Alexiou and Coutts [52] described correlations for sRPR and three HR-based methods by training mode (all *p* < 0.05): technical (r = 0.68 to 0.82), conditioning (r = 0.60 to 0.79), and speed sessions (r = 0.61 to 0.79). Campos-Vazquez et al. [50] also found a large and very large relations between internal methods: HR > 80% HR_max_ and HR > 90% HR_max_–sRPE during ball-possession games, technical and tactical training (r = 0.61 to 0.68); Edward’s TRIMP–sRPE and between TRIMP_MOD_–sRPE in sessions with ball-possession games, technical and tactical training (r = 0.73 to 0.87). The reported correlations between the different HR-based methods were always documented (r = 0.92 to 0.98). These results provide clear evidence about the applicability of HR-based methods and sRPE to measure internal load during various training modes. However, the interchangeable application of these methods to measure load and intensity should consider the low validity to quantify neuromuscular load. Kelly et al. [70] verified correlations within playing positions (WD, r = 0.81; CD, r = 0.74; WM, r = 0.70; CM, r = 0.70; attacker, r = 0.84; *p* < 0.001). The high magnitude of the correlations (large and large to very large) may reflect the lack of specific training for the playing position.

### 4.4. Study Limitations and Future Directions

There are some limitations that should be addressed in the practical application of this review: (i) the different methodological approaches in the reviewed studies; (ii) the related training load measures, metrics, and thresholds that have varied according to the authors; and (iii) methodological constraints about screening procedures. First, several authors point out that the most contentious limitations were the external validity of data collection. Second, future investigations should consider a meta-analytic procedure to quantify training and match load, with which data extrapolation may underestimate the daily and accumulated load. Thirty, we have considered only full-text articles available in English; this was a language limitation in the literature search strategy.

The wide range of sample sizes (9 ≤ *n* ≤160) can influence the data comparability such as the characteristics of observations: monitoring periods (3–43 weeks), total training sessions (17–2981) and training sessions per week (3–6). Moreover, future training load analysis should be focused on different coaches, tapering strategies, and continuous seasons. This longitudinal design might include different teams and competitive levels. In the present review, only two studies compared the accumulated training load performed by two teams [88,89]. Most studies conducted the analysis in only one team and/or club wherefore the data may not be representative to other teams and countries. The included studies adhered to different competitive levels, geographic locations, and populations. More studies are needed in order to obtain greater precision in quantifying training load in different locations and competitive levels. Comparing main differences according to the competitive level can provide important information about the level and experience of the players. The studies recruited adult (*n* = 23) and young (*n* = 7) players as participants. There is a research gap on female players, given that only one study was conducted in this population [52]. Furthermore, there is no evidence-based study in the daily and accumulated load in goalkeepers, except an exploratory case study that provide a physical load report during a competitive season [123,124].

Several studies included GPS systems in their procedures with different specifications and sampling frequencies (i.e., 5, 10, and 15 Hz). The validity and reliability were well documented in the literature [25,26]. There were some limitations when applying a sampling frequency between 1 Hz and 5 Hz during distances covered at high intensity, speed-based measures, and short linear distances with changes of direction [112]. GPS devices at 10 Hz seem to be the most valid and reliable systems whereas the increase in sampling frequency to 15 Hz does not seem to provide any additional benefit assessing team sport movements [125,126]. The concurrent use of a tracking system (i.e., GPS or LPM) and semi-automatic multiple-camera system (i.e., Prozone^®^) to quantify training and match demand has obvious implications for the data comparability. The integration of different tracking systems is a methodological strategy applied in three reviewed articles [70,87,106] but there is a moderate typical error in this kind of estimation [10].

The use of GPS technology to estimate energy expenditure during the training session may be underestimated [106]. Metabolic power variables seem to be more suitable to determine high-intensity movements than estimations based on speed [90]. The importance of including acceleration and accelerometer variables to quantify external load was well documented in the present systematic review. The accelerometer parameters including body impact, body load, player load or dynamic-stress load, and the acceleration and deceleration were supported in several reviewed studies. There was no consensus on the use of acceleration thresholds [53]. In addition, the comparison between acceleration variables measured with different tracking systems would be challenging [10]. Future research should focus on comparing demands for acceleration between training sessions and official matches-play measured with the same tracking system. Moreover, specific playing position should be taken into consideration.

The daily and accumulated load were usually lower than other team sports (e.g., Australian Football) and endurance sports [70]. The reviewed studies reported an intra-week variation and gradual reduction to MD-1 or MD-2, which means coaches’ staff reduced volume and intensity during training sessions as competition approached. However, the majority of studies failed to provide any specific context associated with each training day, which may limit the application of such data. None of the reviewed studies focused on training and match load analysis in the seasonal variations during specific training interventions. Therefore, it would be interesting to discover what training modes, sub-components, and exercise typologies contribute to increases or decreases (fluctuations) in certain load measures [127,128]. In team sports (in this case, football), there are some methodological challenges in training load quantification. However, it is not possible to argue that there was a direct causal relationship between physical performance and/or team performance. The dynamic and unpredictable nature of match-play may make it impossible to adjust training and matches [95]. This fact limited the understanding of the relationship between training periodization and individual and team performance [129].

The loading discrepancies within playing positions may significantly affect individual performance and increase injury risk [33]. Therefore, quantifying training load can adjust training periodization models and individualized training sessions. Additionally, Owen et al. [99] allowed the possibility of describing the daily and positional variations through the multi-modal mechanical approach. The content and magnitude of the complementary training sessions were not reported in the literature; wherefore, future investigations about training mode or sub-components effect are recommended. Martin-Garcia et al. [82] noted that future studies should implement mixed small-sided games and running exercise strategies to infer the greatest training stimulus for players with limited playing time (i.e., non-starters and/or fringe player).

The training load quantification in youth football suggested that as players grow older, the training process focus moves to competition whereas in younger players, the training goals were physical and technical development [47,48,60]. Therefore, the weekly microcycle should be adjusted for age. The influence of different weekly microcycle schedules has not yet been established for the competitive performance and long-term development of youth football players. The youth training responses differed markedly from adult and professional players’ due to the development stage of sport specified-skills and physical attributes [130].

The relationships between internal and external load should be interpreted with regard to some limitations. According to Impellizzeri et al. [54], only 50% of HR loading variation were supported by sRPE. However, there is a limitation inherent in the use of HR-based measures to quantify training intensity during anaerobic efforts. This fact may influence the magnitude of the correlation between perceived exertion and HR loading. The perceived exertion appears to be better linked with external load when the speed zones were individually determined than when the arbitrary speed zones [107]. Notwithstanding, there were some limitations in the achievement of real individual maximal values (e.g., maximal aerobic speed) and the speed zone transition. The speed zone transition is very important due to the significant physiological effort associated [131].

## 5. Conclusions

The present systematic review provided the first report about monitoring accumulated training and match load in football players. Current research suggests that the training and match load variation seem to be influenced by the type of weekly schedule, player’s starting status, playing positions, age group, training mode, and contextual factors. Therefore, there was a related high variation in the weekly loading distribution and a limited load variation during a competitive season. Most of the evidence has implications for adult male professional football players concerning the large body of quantitative studies (QS: 0.65–0.89). In youth football, the studies appear to indicate a small fluctuation across weekly and seasonal accumulated load. However, further studies are recommended to improve knowledge on the female and youth accumulated training/match load monitoring.

## Figures and Tables

**Figure 1 ijerph-18-03906-f001:**
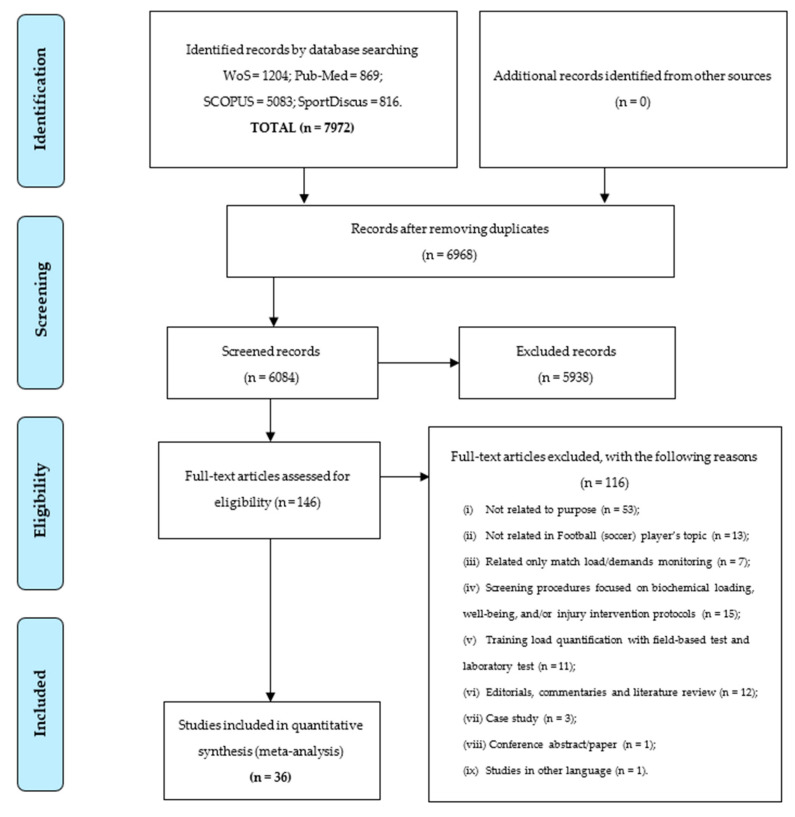
Preferred reporting item for systematic reviews and meta-analyses (PRISMA) flow diagram.

**Table 1 ijerph-18-03906-t001:** Search terms and following keywords for screening procedures.

Search Term		Keywords
Football (population)	1	*(“football” OR “soccer” OR “association football”)*
Training load (dependent variable)	2	*(“training load” OR “external training load” OR “workload” OR “internal training load” OR “external load” OR “internal load”)*
Periodization (independent variables)	3	*(“periodization” OR “schedule” OR “distribution” OR “week” OR “microcycle” OR “mesocycle” OR “season phase”) AND (“in-season” OR “pre-season” OR “preparation” OR “off-season” OR “post-season”)*
Boolean search phrase (final search)	4	*1 AND (2 OR 3)*

**Table 2 ijerph-18-03906-t002:** Summary of measure and measurements in the included articles.

Construct	Measure	Measurement	Thresholds and/or Metric Formula	Reference	Further Reading
Internal Load	Heart Rate	% HR_max_	Zone 1: ≤75% HR_max_; zone 2: 75–84.9% HR_max_; zone 3: 85–89.9% HR_max_; zone 4: ≥90% HR_max_.	[47,48]	[49]
			Zone 1: ≤75% HR_max_; zone 2: 75–84.9% HR_max_; zone 3: 85–89.9% HR_max_; zone 4: ≥90% HR_max_.	[50]	[51]
			Zone 1: 50–60% HR_max_; zone 2: 60–70% HR_max_; zone 3: 70–80% HR_max_; zone 4: 80–90% HR_max_; zone 5: 90–100% HR_max_.	[52,53,54,55,56,57,58,59,60,61]	[62,63,64]
		LT_zone_	zone 1: <LT; zone 2: between LT and AT; zone 3: >AT (k = 1 for zone 1; k = 2 for zone 2, and k = 3 for zone 3)	[52,54]	[63]
		Bannister TRIMP	D × (∆HR_ratio_) × (0.64 × e ^b × HRB^) (D = (∆HR_ratio_) [(HR_TS_– HR_B_)/(HR_max_– HR_B_)]) weighting factor (k) = 1.62 (females);1.92 (males)	[50,52,54,57,58]	[63]
		Edward’s TL	D (zone 1) × 1+ D (zone 2) × 2 + D (zone 3) × 3+ D (zone 4) × 4 + D (zone 5) × 5	[52,54,57,65]	[66]
		Lucia’s TL/LT_zone_ TL	D (zone 1) × 1+ D (zone 2) × 2 + D (zone 3) × 3	[52,54]	[66,67]
		Stagno TL/TRIMP_MOD_	[(HR_TS_ − HR_B_)/(HR_max_ − HR_B_)]) weighting factor = 0.1225e^3.9434x^	[50]	[51]
		HR-TL	∑ (time (min) spent in zone × numerical factor of zone)	[56]	[68]
	Perceived Exertion	sRPE	RPE × D	[52,54,55,56,57,58,59,69,70,71,72]	[64,73,74,75]
		sRPEresp TL/sRPEmusc TL	sRPE × D	[71]	[76,77]
		Fatigue score	Seven-point scale: training exertion, sleep quality, muscle soreness, infection/illness, concentration, training efficiency, anxiety/irritability, and general stress.	[69]	[78]
		HI	Fatigue, stress muscle soreness, and quality sleep.	[72]	[79]
External load	Distance and speed	Speed zones/ thresholds	Zone 1: 0–6.9 km× h^−1^; zone 2: 7.0–9.9 km × h^−1^; zone 3: 10.0–12.9 km × h^−1^; zone 4: 13–15.9 km × h^−1^; zone 5: 16–17.9 km × h^−1^; and zone 6: ≥18.0 km × h^−1^ (sprints).	[47,48]	[80,81]
			Walking/jogging: <10.8 km × h^−1^; HSR: ≥20.9 km × h^−1^; SPR: >24.1 km × h^−1^.	[53,82,83]	[32,40,84,85,86]
			Standing: 0–0.6 km × h^−1^; walking: 0.7–7.1 km × h^−1^; jogging: 7.2–14.3 km × h^−1^; running: 14.4–19.7 km × h^−1^; HSR: 19.8–25.1 km × h^−1^; SPR: >25.1 km × h^−1^.	[87,88,89,90,91]	[92,93]
			Running: 11.4–18.9 km × h^−1^; HSR: 15.0–18.9 km × h^−1^; SPR: >19.0 km × h^−1^.	[58]	[94]
			Walking: 0–6.9 km × h−1; jogging: 7.0–13.9 km × h^−1^; Running: 14.0–20.0 km × h^−1^; SPR: >20.0 km × h^−1^.	[88,89,95]	[96]
			Low-speed running: <14.4 km × h^−1^; HSR: >19.8; SPR: >25.2 km × h^−1^.	[57,70,91,97,98,99]	[11,100,101,102,103,104]
			Low-speed running: <14 km × h^−1^; HSR: 14.4 km × h^−1^; HSR: 19.8–25.2 km × h^−1^	[102,105]	[104]
			HSR: >19 km × h^−1^.	[72]	[104]
			HSR: >16 km × h^−1^.	[61]	[85]
			Standing/walking: 0–7.2 km × h^−1^; low intensity running: 7.3–14.3 km × h^−1^; moderate intensity running: 14.4–21.5 km × h^−1^; HSR: 19.8–25.1 km × h^−1^; very HSR > 25.1 km × h^−1^.	[106,107,108]	[30,40,109]
	Acceleration	Acceleration zones/ thresholds	Low: 1–2 m × s^−2^; Moderate: 2–3 m × s^−2^; High: >3 m × s^−2^.	[53]	[85]
			ACC: >2.5 m × s^−2^; DEC: <2.5 m × s^−2^.	[91]	[40]
			ACC: >2 m × s^−2^.	[99,110]	[111]
			ACC/DEC: >3 m × s^−2^.	[61,82,98]	[86]
			ACC: >4 m × s^−2^.	[108]	[100]
			ACC: medium (1.5–3.0 m × s^−2^); high (>3.0 m × s^−2^). DEC: medium (−1.5 to −3.0 m × s^−2^); high (<−3.0 m × s^−2^).	[82]	[102,111]
	Accelerometry	Body impacts/body load	Zone 1: 5.0–6.0 g; zone 2: 6.1–6.5 g; zone 3: 6.5–7.0 g; zone 4: 7.1–8.0 g; zone 5: 8.1–10.0 g; zone 6: ≥10.1 g.	[47,48,98,99]	[112,113,114]
		Player load	$\sqrt{{\left(a_{x1}\;-\;a_{y-1}\right)}^{2}+(a_{y1}\;-\;a_{y-1}{)}^{2}+{\left(a_{z1}\;-\;a_{z-1}\right)}^{2}\;}/100$	[53,88,89,115]	[10,116]
		Player load	${\left(a_{x1}\;-\;a_{y-1}\right)}^{2}+(a_{y1}\;-\;a_{y-1}{)}^{2}+{\left(a_{z1}\;-\;a_{z-1}\right)}^{2}$	[57]	[117]
		Dynamic-stress load	∑ (body load for each zone × body mass)	[118]	[119]
Ratios/ scores	Ratio/scores (Weekly TL)	Work: rest ratio	High to very high: >16 km × h^−1^; moderate: 10.0–15.9 km × h^−1^; low intensity: 7.0–9.9 km × h^−1^; very low intensities: 0–6.9 km × h^−1^ (normalized for each 100 m).	[47]	[80]
		Work: rest ratio	WRR: distance covered at a speed ≥ 4 km × h^−1^ period of activity or work divided by the distance covered at a speed <3.9 km × h^−1^; period of recovery or rest); FEHS ≥ 18 km × h^−1^; FESS ≥ 21 km × h^−1^.	[115]	[119]
		THIA (%)	∑ (MSR, HSR and SPR)	[107]	originally proposed by Rago et al. [107]
	Ratio/scores (Weekly TL and ML)	TMr	(Weekly load)/(Match load)	[95]	originally proposed by Clemente et al. [95]
		Session volume score	$\left({\mathrm{Volume}\;\mathrm{Metric}\;x}_{1},x_{2},x_{3},x_{4}\;\mathrm{of}\;\mathrm{MD}\;(\%)/4\right)$	[97]	originally proposed by Owen et al. [97]
		Session intensity score	$\left({\mathrm{Intensity}\;\mathrm{Metric}\;x}_{1},x_{2},x_{3},x_{4}\mathrm{of}\;\mathrm{MD}\;(\%)/4\right)$	[97]	originally proposed by Owen et al. [97]
Energy cost and metabolic power	Equivalent- estimation	EC	EC = 155.4 × 155.4 × ES^4^ × 155.4 × ES^3^ × 155.4 × ES^2^ × 155.4 × ES × EM × KT	[90]	[84,85]
		P_met_	HP: 20–35 W× kg^−1^; EP: 35–55 W × kg^−1^;: >55 W × kg.	[82,83,118]	[84,85]

∆HR—HR variation; ACC—acceleration; AT—anaerobic threshold; D—duration; DEC—deceleration; EC—energy cost; EM—equivalent body mass; EP—elevated power; ES—equivalent slope; FEHS—frequency of efforts at high speed (≥18 km × h^−1^); FESS—frequency of efforts at sprint speed (≥21 km × h^−1^); HI—Hooper Index; HP—high power; HR—heart rate; HR_B_—basal heart rate; HR_max_—maximum heart rate; HR_TL_—heart rate training load; HR_TS_—average training session heart rate; HSR—high speed running; K—coefficient relative; KT—constant; LT_zone_—lactate threshold; LT_zone_ lactate threshold zone; MD—match day; ML—match load; MS—maximum power; P_met_—equivalent metabolic power; RPE—ratings of perceived exertion; SPR—sprinting; sRPE—sessions ratings of perceived exertion; sRPEmusc-TL—sessions ratings of muscular training load; sRPEres-TL—sessions ratings of respiratory training load; THIA—total high-intensity activity; TL—training load; TMr—training/match ratio; TRIMP—training impulse; TRIMP_MOD_—modified training impulse; WRR—work:rest ratio.

**Table 3 ijerph-18-03906-t003:** Summary characteristics of the participants’ demographics recruited in the studies included in the systematic review and its quality score.

Reference (Year)	Study Design	Population	Competitive Level, Country	Sample (N)	Sex	Age (y)	Stature	Body Mass (kg)	QS
Abade et al. [47]	Prospective Cohort	Youth	Elite, Portugal	151	Male	U15 (*n* = 56): 14.0 ± 0.2 U17 (*n* = 66): 15.8 ± 0.4 U19 (*n* = 29): 17.8 ± 0.6	U15 (*n* = 56): 1.71 ± 0.07 U17 (*n* = 66): 1.76 ± 0.06 U19 (*n* = 29): 1.77 ± 0.07	U15 (*n* = 56): 60.1 ± 6.3 U17 (*n* = 66): 65.8 ± 5.5 U19 (*n* = 29): 70.0 ± 5.6	0.78
Akenhead et al. [53]	Prospective Cohort	Adult	Elite, UK	33	Male	24.0 ± 4.0	1.83 ± 0.05	82 ± 8.0	0.87
Alexiou and Coutts [52]	Prospective cohort	Adult	Elite, Portugal	15	Female	19.3 ± 2.0	1.69 ± 0.05	64.8 ± 7.7	0.83
Anderson et al. [87]	Prospective Cohort	Adult	Elite, UK	12	Male	25.0 ± 5.0	1.80 ± 0.05	81.5 ± 7.5	0.78
Anderson et al. [106]	Prospective Cohort	Youth	Elite, UK	19	Male	25.0 ± 4.0	1.78 ± 0.06	80.6 ± 8.3	0.74
Baptista et al. [97]	Prospective Cohort	Adult	Elite, Norway	18	Male	ND	ND	ND	0.74
Brito et al. [69]	Prospective Cohort	Adult	Elite, France	13	Male	18.6 ± 0.5	1.77 ± 0.05	70.0 ± 7.3	0.78
Campos-Vazquez et al. [50]	Prospective Cohort	Adult	Elite, Spain	9	Male	26.7 ± 4.5	1.77 ± 0.07	74.5 ± 5.7	0.74
Casamichana et al. [115]	Prospective Cohort	Adult	Elite, Spain	28	Male	22.9 ± 4.2	1.77 ± 0.05	73.6 ± 4.4	0.87
Clemente et al. [89]	Prospective Cohort	Adult	Elite, Portugal and The Netherlands	29	Male	PT (*n* = 14): 19.21 ± 1.05 NL (*n* = 15): 25.14 ± 3.90	PT (*n* = 14): 1.80 ± 0.06 NL (*n* = 15): 1.79 ± 0.06	PT (*n* = 14): 74.07 ± 6.21 NL (*n* = 15): 73.21 ± 6.46	0.74
Clemente et al. [95]	Prospective Cohort	Adult	Elite, Portugal	27	Male	24.9 ± 3.5	1.69 ± 0.41	71.6 ± 18.7	0.83
Clemente et al. [88]	Prospective Cohort	Youth	Elite, Portugal and The Netherlands	89	Male	NL1 (*n* = 18): 25.39 ± 4.82 NL2 (*n* = 24): 21.46 ± 2.50 NL3 (*n* = 23): 23.00 ± 3.70 PT (*n* = 24): 24.70 ± 2.92	NL1 (*n* =18):1.84 ± 0.05 NL2 (*n* = 24):1.80 ± 0.08 NL3 (*n* = 23):1.84 ± 0.06 PT (*n* = 24): 1.81 ± 0.06	NL1 (*n* = 18): 77.29 ± 4.73 NL2 (*n* = 24): 71.73 ± 8.61 NL3 (*n* = 23): 74.50 ± 6.90 PT (*n* = 24): 77.48 ± 6.80	0.87
Clemente et al. [105]	Prospective Cohort	Adult	Elite, Europe *	19	Male	26.5 ± 4.3	1.80 ± 7.3	75.6 ± 9.6	0.83
Coutinho et al. [47]	Prospective Cohort	Adult	Elite, Portugal	151	Male	U15 (*n* = 2 56): 14.0 ± 0.2 U17 (*n* = 66): 15.8 ± 0.4 U19 (*n* = 29): 17.8 ± 0.6	U15 (*n* = 56): 1.71 ± 0.07 U17 (*n* = 66): 1.76 ± 0.06 U19 (*n* = 29): 1.77 ± 0.07	NL1 (*n* = 18): 77.29 ± 4.73 NL2 (*n* = 24): 71.73 ± 8.61 NL3 (*n* = 23): 74.50 ± 6.90 PT (*n* = 24): 77.48 ± 6.80	0.74
Dalen et Lorås [102]	Prospective Cohort	Youth	Pre-Elite, Norway	18	Male	15.7 ± 0.5	1.78 ± 4.6	67.1 ± 5.5	0.83
Gaudino et al. [90]	Prospective Cohort	Adult	Elite, UK	26	Male	26.0 ± 5.0	1.82 ± 0.07	79.0 ± 5.0	0.78
Gaudino et al. [118]	Prospective Cohort	Youth	Elite, UK	22	Male	26.0 ± 6.0	1.82 ± 0.07	79.0 ± 7.0	0.74
Impellizzeri et al. [54]	Prospective Cohort	Adult	ND	19	Male	17.6 ± 0.7	1.79 ± 0.05	70.2 ± 4.7	0.87
Jeong et al. [55]	Prospective Cohort	Adult	Elite, Korea	20	Male	24.0 ± 3.0	1.78 ± 0.06	73.0 ± 4.0	0.78
Kelly et al. [70]	Prospective Cohort	Adult	Elite, UK	111	Male	27.0 ± 5.4	1.81 ± 0.07	77.0 ± 6.6	0.78
Kelly et al. [65]	Prospective Cohort	Youth	Elite, UK	26	Male	27.0 ± 5.4	1.81 ± 0.07	77.0 ± 6.6	0.83
Los Arcos et al. [71]	Prospective Cohort	Adult	Elite, Spain	24	Male	20.3 ± 2.0	1.79 ± 0.05	73.0 ± 5.6	0.74
Malone et al. [56]	Prospective Cohort	Adult	Elite, UK	30	Male	25.0 ± 5.0	1.83 ± 0.07	80.5 ± 7.4	0.70
Martin-Garcia et al. [82]	Prospective Cohort	Adult	Elite, Spain	24	Male	20.0 ± 2.0	1.78 ± 0.64	70.2 ± 6.1	0.78
Marynowicz et al. [98]	Prospective Cohort	Youth	Elite, ND	18	Male	17.1 ± 0.96	1.79 ± 4.77	70.9 ± 4.7	0.83
Oliveira et al. [72]	Prospective Cohort	Adult	Elite, ND	19	Male	26.3 ± 4.3	1.84 ± 0.07	78.5 ± 6.8	0.89
Owen et al. [108]	Prospective Cohort	Adult	Elite, ND	16	Male	26.7 ± 4.07	1.83 ± 0.06	78.4 ± 8.03	0.74
Owen et al. [99]	Prospective Cohort	Adult	Elite, Swiss	29	Male	26.7 ± 4.0	1.83 ± 0.06	78.4 ± 8.0	0.83
Rago et al. [107]	Prospective Cohort	Adult	Elite, Italy	13	Male	25.8 ± 3.5	1.82 ± 0.06	78.3 ± 5.9	0.87
Rago et al. [61]	Prospective Cohort	Adult	Elite, Spain	23	Male	27.8 ± 3.9	1.78 ± 6.4	72.7 ± 11.9	0.87
Sanchez-Sanchez et al. [91]	Prospective Cohort	Adult	Amateur, Brazil	160	Male	20.8 ± 1.7	1.76 ± 0.04	69.7 ± 2.9	0.65
Scott et al. [57]	Prospective Cohort	Adult	Elite, Australian	15	Male	24.9 ± 5.4	1.81 ± 0.07	77.6 ± 7.5	0.74
Swallow et al. [110]	Prospective Cohort	Adult	Pre-Elite, UK	24	Male	26.0 ± 6.0	1.81 ± 8.0	79.7 ± 7.8	0.74
Stevens et al. [83]	Prospective Cohort	Youth	Elite, The Netherlands	28	Male	21.9 ± 3.2	1.82 ± 0.07	76 ± 7.0	0.83
Vahia et al. [58]	Prospective Cohort	Youth	Elite, UK	15	Male	16.7 ± 1.0	1.76 ± 0.05	69.9 ± 6.9	0.74
Wrigley et al. [60]	Prospective Cohort	Youth	Elite, UK	24	Male	U14 (*n* = 8): 13.0 ± 1.0 U16 (*n* = 8): 15.0 ± 1.0 U18 (*n* = 8): 17.0 ± 1.0	U14 (*n* = 8): 1.61 ± 0.06 U16 (*n* = 8): 1.74 ± 0.07 U18 (*n* = 8): 1.79 ± 0.05	U14 (*n* = 8): 48.0 ± 10.3 U17 (*n* = 66): 67.3 ± 8.1 U19 (*n* = 29): 73.5 ± 4.4	0.78

kg—kilogram (SI); m—meters (SI); ND—not described; NL—The Netherlands; PT—Portugal; UK—United Kingdom; U14—under-14; U15—under-15; U16—under-16; U17—under-17; U18—under-18; U19—under-19; QS—quality score. Note: * Country is not specified.

**Table 4 ijerph-18-03906-t004:** Methodological approaches of included articles.

Reference (Year)	Observations Sample					Training Load Measures/Metrics		Device Specification (Manufacturer Model and Specs)	
	Monitoring Period	Training Sessions	TS/Week	Sets	Match-Play	Internal Load	External Load	Internal Load	External Load
Abade et al. [47]	9 weeks	38 TS	4 TS/wk (~90 min)	612	ND	**HR:** %HR_max_	**Distance and speed:** TD covered (m); relative distance or pace (m × min^−1^); D in different speed zone (km × h^−1^); and sprints (number and time interval). **Accelerometry:** absolute and relative body impacts (g).	5 Hz short-range telemetry system (Polar Team System, Kempele, Finland).	15 Hz GPS and 100-Hz MEMS (SPI-Pro X II, GPSports, Canberra, Australia).
Akenhead et al. [53]	12 weeks	48 TS	5 TS/wk	295	1 MP/wk (90 min)	**HR:** %HR_max_	**Distance and speed:** TD covered (m); HSR (km × h^−1^); and SPR (km × h^−1^). **Acceleration:** ACC_TOTAL_ (m × s^−2^) and DEC_TOTAL_ (m × s^−2^). **Accelerometry:** PL (g).	1 Hz short-range telemetry system (Team 2, Polar Electro, Kempele, Finland).	10 Hz GPS and 100-Hz MEMS (Catapult MiniMaxx S4, Firmware 6.7, Melbourne, Australia).
Alexiou and Coutts [52]	16 weeks	623 TS	ND	ND	623 MP	**HR:** Bannister TL, Edward’s TL and LTzone TL.	ND	1 Hz short-range telemetry system Polar NV, Polar Electro, Kempele, Finland).	ND
Anderson et al. [87]	3 weeks	10 TS	5 TS/wk	145	6 MP	ND	**Distance and speed:** TS duration (min); TD covered (m); AvS (m × min^−1^); and D in different speed zones (km × h^−1^).	ND	10 Hz GPS (Viper pod 2, STATSports^®^, Newry, Northern Ireland) and semi-automatic multiple-camera system (Prozone Sports Ltd., Leeds, United Kingdom).
Anderson et al. [106]	39 weeks	181 TS	ND	2182	7 MP	ND	**Distance and speed:** TS duration (min); TD covered (m); and D in different speed zones (km × h^−1^).	ND	10 Hz GPS (Viper pod 2, STATSports, Northern Ireland) and semi-automatic multiple-camera system (Prozone Sports Ltd.^®^, Leeds, United Kingdom).
Baptista et al. [97]	11 weeks	537	4 TS/wk	630	15 M	ND	**Distance and speed:** TD covered (m); HSR_peak_ (km × h^−1^); and SPR_peak_ (km × h^−1^). **Acceleration:** ACC_peak_ (m × s^−2^) and DEC_peak_ (m × s^−2^).	ND	Stationary radio-based tracking system (ZXY Sport Tracking System, Trondheim, Norway)
Brito et al. [69]	36 weeks	2591 TS	5–11 TS/wk	ND	ND	**Perceived Exertion:** RPE, sRPE, and perceived fatigue.	ND	CR10 and fatigue questionnaire.	ND
Campos-Vazquez et al. [50]	ND	ND	5 TS/wk (~90 min)	ND	ND	**HR:** Edwards TL and Stagno TL/TRIMP_MOD_. **Perceived Exertion**: RPE and sRPE.	ND	CR10 and 1 Hz short-range telemetry system (Team 2, Polar Electro, Kempele, Finland).	ND
Casamichana et al. [115]	ND	44 TS	2/3 TS/wk (~90 min)	ND	ND	ND	**Distance and speed:** TD covered (m); DHS (km × h^−1^); and DSS (km × h^−1^). **Accelerometry:** PL (g). **Ratios/scores:** WRR (km × h^−1^); FEHS (km × h^−1^); and DHS (km × h^−1^).	ND	10 Hz GPS and 100-Hz MEMS (Catapult MinimaxX Team Sport 4.0, Melbourne, Australia).
Clemente et al. [89]	ND	44 TS	3 TS/wk (~90 min)	ND	ND	ND	**Distance and speed:** TD covered (m); relative distance covered or pace (m/min); D in different speed zones; maximum speed (km × h^−1^); and number of sprints per minute (n × min^−1^). **Accelerometry:** PL (g).	ND	10 Hz GPS and 100-Hz MEMS (JOHAN Sports, Noordwijk, The Netherlands).
Clemente et al. [95]	5 weeks	ND	5 TS/wk	ND	ND	ND	**Distance and speed:** TD covered (m); relative distance covered or pace (m/min); D in different speed zones; maximum speed (km × h^−1^); and number of sprints per minute (n × min^−1^). **Acceleration:** ACC (m × s^−2^) and DEC (m × s^−2^). **Accelerometry:** PL (g). **Ratios/scores:** TMr.	ND	10 Hz GPS and 100-Hz MEMS (JOHAN Sports, Noordwijk, The Netherlands).
Clemente et al. [88]	7 weeks	ND	5–6 TS/wk	ND	ND	ND	**Distance and speed:** TD covered (m); relative distance covered or pace (m/min), D in different speed zones; maximum speed (km × h^−1^); and number of sprints per minute (n × min^−1^). **Accelerometry:** PL (g).	ND	10 Hz GPS and 100-Hz MEMS (JOHAN Sports, Noordwijk, The Netherlands).
Clemente et al. [105]	45 weeks	197 TS	ND	ND	44 MP	ND	**Distance and speed:** TD covered (m); relative distance covered or pace (m × min^−1^), D in different speed zones; maximum speed (km × h^−1^); and number of sprints per minute (n × min^−1^).	ND	18-Hz MEMS and 100-Hz tri-axial accelerometer (STATSports, Apex, Newry, Northern Ireland).
Coutinho et al. [47]	22 weeks	ND	3–4 TS/wk	ND	ND	**HR:** %HRmax	**Distance and Speed:** TD covered (m); relative distance or pace (m × min^−1^); D in different speed zone (km × h^−1^); and sprints (number and time interval). **Accelerometry:** Absolute and relative body impacts (g).	5 Hz short-range telemetry system (Polar Team System, Polar, Kempele, Finland).	15 Hz GPS and 100-Hz MEMS (SPI-Pro X II, GPSports, Canberra, Australia)
Dalen et Lorås [102]	10 weeks	38 TS	4 TS/wk	ND	10 MP	**HR:** Banister TL	**Distance and speed:** TD covered (m); relative distance covered or pace (m × min^−1^); D in different speed zones; and maximum speed (km × h^−1^). **Acceleration:** ACC (m × s^−2^) and DEC (m × s^−2^).	5 Hz short-range telemetry system (Polar Team System, Polar, Kempele, Finland)	10 Hz and 100-Hz MEMS (Polar Team System, Polar, Kempele, Finland).
Gaudino et al. [90]	10 weeks	628 TS	24 TS/player	ND	ND	ND	**Distance and speed:** D in different speed zone (km × h^−1^). **Energy and metabolic power:** P_met_ (W × kg^−1^) and metabolic load distance (W × kg^−1^).	ND	15 Hz GPS and 100-Hz MEMS (SPI-Pro X II, GPSports, Canberra, Australia).
Gaudino et al. [118]	38 weeks	1892 TS	3–4 TS/wk (~60 min)	ND	ND	**Perceived Exertion:** RPE and sRPE.	**Distance and speed:** D in different speed zone (km × h^−1^). **Energy and metabolic power:** P_met_ (W × kg^−1^) and metabolic load distance (W × kg^−1^). **Accelerometry:** dynamic-stress load (AU).	CR10	10 Hz GPS and 100-Hz MEMS (Viper Pod, STATSports, Newry, Northern Ireland)
Impellizzeri et al. [54]	9 weeks	479 TS	3–4 TS/wk (~60 min)	ND	ND	**HR:** Edwards TL, Banister TL, and Lucia TL. **Perceived Exertion:** RPE and sRPE.	ND	CR10 and 5 Hz short-range telemetry system (VantageNV, Polar Electro, Kempele, Finland).	ND
Jeong et al. [55]	10 weeks	628 TS	24 TS/players (~60 min)	ND	6 MP	**HR:** %HR_max_. **Perceived Exertion:** RPE and sRPE	ND	CR10 and 5 Hz short-range telemetry system (Polar Team System, Polar, Kempele, Finland).	ND
Kelly et al. [70]	36 weeks	ND	ND TS/wk (~60 min)	ND	49 MP	**Perceived Exertion:** RPE and sRPE	**Distance and speed:** TD covered (m); and D in different speed zones (km × h^−1^).	CR 10	10 Hz GPS (SPI-Pro X II, GPSports, Canberra, Australia) and semi-automatic multiple-camera system (Prozone Sports Ltd.^®^, Leeds, United Kingdom).
Kelly et al. [65]	43 weeks	1010 TS	55 TS/player	ND	ND	**HR:** %HR_max_. **Perceived Exertion:** RPE and sRPE	ND	CR 10	ND
Los Arcos et al. [71]	35 weeks	ND	4–5 TS/wk (~90–104 min)	ND	ND	**Perceived Exertion:** sRPEres-TL and sRPEmus-TL	ND	CR 10	ND
Malone et al. [56]	7 weeks	27 TS	3–4 TS/wk	ND	ND	**HR:** %HR_max_ **Perceived Exertion:** RPE and sRPE	ND	CR10 and Portable team-based HR receiver (Acentas GmBH^®^, Freising, Germany; Firstbeat Sports, Jyväskylä, Finland)	15 Hz GPS and 100-Hz MEMS (SPI-Pro X II, GPSports, Canberra, Australia)
Martin-Garcia et al. [82]	12 weeks	17 TS	5 TS/wk (~83–92 min)	ND	ND	ND	**Distance and speed:** TS duration (min); TD covered (m); and D in different speed zones (km × h^−1^). **Acceleration:** ACC (m × s^−2^) and DEC (m × s^−2^). **Metabolic power:** AMP power per second and kg (W × kg^−1^); and metabolic load distance (W × kg^−1^).	ND	10 Hz GPS (Viper Pod, STATSports, Canberra, Australia)
Marynowicz et al. [98]	18 weeks	12–76 TS/player	ND	804	ND	**Perceived Exertion:** RPE and sRPE	**Distance and speed:** TD covered (m); relative distance covered or pace (m × min^−1^); D in different speed zones. **Acceleration:** ACC (m × s^−2^) and DEC (m × s^−2^). **Accelerometry:** PL (g).	CR 10	10 Hz GPS and 400 Hz tri-axial accelerometer (Player Tek^TM^, Catapult, Melbourne, Australia).
Oliveira et al. [72]	45 weeks	111 TS	4 TS/wk	ND	1 MP/wk (90 min)	**Perceived Exertion:** RPE, sRPE, and HI.	**Distance and speed:** TS duration (min); TD covered (m); D in different speed zones (km × h^−1^); AvS (m × min^−1^). **Acceleration:** ACC (m × s^−2^) and DEC (m × s^−2^) **Accelerometry:** PL (g) and number of impacts.	CR10	10 Hz GPS (Viper pod 2, STATSports, Newry, Northern Ireland)
Owen et al. [108]	39 weeks	2981 TS	16–20 TS/M	ND	50 MP/season	ND	**Distance and speed:** TD covered (m) and D in different speed zones (km × h^−1^). **Acceleration:** ACC (m × s^−2^) and DEC (m × s^−2^). **Ratios/Scores:** session volume and intensity.	ND	10-Hz GPS (Viper, Statsport, Newry, Northern Ireland)
Owen et al. [99]	42 weeks	490 TS	5 TS/wk (~61–74 min)	ND	37 MP	**Perceived Exertion:** RPE, CR10, and sRPE	**Distance and speed:** TD covered (m); relative distance covered or pace (m × min^−1^), D in different speed zones (km × h^−1^); maximum speed (km × h^−1^); and number of sprints per minute (n × min^−1^).	CR10	10 Hz GPS (Catapult Innovations, Melbourne, Australia).
Rago et al. [107]	6 weeks	24 TS	4 TS/wk	ND	ND	**Perceived Exertion:** RPE and sRPE	**Distance and speed:** TD covered (m); D in different speed zones (km × h^−1^); and THIA (%). **Acceleration:** ACC (m × s^−2^) and DEC (m × s^−2^).	CR 10	10-Hz GPS (BT-Q1000 Ex, QStarz, Taipei, Taiwan)
Rago et al. [61]	~13 weeks	67 TS	ND	828	15 MP	**HR:** %HR_max._	**Distance and speed:** TD covered (m) and D in different speed zones (km × h^−1^) **Acceleration:** ACC (m × s^−2^) and DEC (m × s^−2^).	5 Hz short-range telemetry system (WIMU PRO; RealTrack Systems SL, Almería, España).	10-Hz GPS with Triaxial accelerometer (WIMU PRO; RealTrack Systems SL, Almería, España)
Sanchez-Sanchez et al. [91]	8 weeks	42 TS	5 TS/wk (~75–120 min)	ND	ND	ND	**Distance and speed:** TS duration (min); TD covered (m); and D in different speed zones (km × h^−1^). **Acceleration:** ACC (m × s^−2^) and DEC (m × s^−2^).	ND	10 Hz GPS (K-GPS, Montelabbate, Italy)
Scott et al. [57]	20 weeks	97 TS	4 TS/wk	ND	1 MP/wk (90 min)	**HR:** Edwards and Banister TL **Perceived Exertion: ** RPE, CR10, and sRPE.	**Distance and speed:** TD covered (m) and D in different speed zones (km × h^−1^). **Accelerometry:** PL (g).	CR10 and 5 Hz short-range telemetry system (Polar Team System, Polar, Kempele, Finland).	5 Hz GPS (Catapult Firmware 6.59, Innovations, Scoresby, Australia) and tri-axial accelerometer (Kionix: KXP94)
Swallow et al. [110]	ND	1029 TS	ND	ND	3–55 MP	ND	**Distance and speed:** TS duration (min); TD covered (m); and D in different speed zones (km × h^−1^). **Acceleration:** ACC (m × s^−2^) and DEC (m × s^−2^). **Accelerometry:** PL (g)**.**	ND	5 Hz GPS and 100 Hz tri-axial accelerometer (Player Tek^TM^, Catapult Cloud, Catapult Sports Group, Australia).
Stevens et al. [83]	33 weeks	ND	3 TS/wk	536	1/2 MP/wk	**HR:** %HR_max_.	**Distance and speed:** TD covered (m) and D in different speed zones (km × h^−1^). **Acceleration:** ACC (m × s^−2^). **Accelerometry:** PL (g)	LPM-integrated Polar Wearlink^®^ technology (Polar Electro Oy, Kempele, Finland).	LPM system (version 05.91 T; Inmotiotec GmbH, Regau, Austria).
Vahia et al. [58]	~30 weeks	1029 TS	4 TS/wk	ND	3 MP	**HR:** Edwards and Banister TL. **Perceived Exertion:** RPE, sRPE.	ND	CR10 and 1 Hz short-range telemetry system (Team 2, Polar Electro, Kempele, Finland).	ND
Wrigley et al. [60]	~30 weeks	160 TS	7 TS/wk	612	1 MP/wk	**HR:** %HR_max_. **Perceived Exertion:** RPE and sRPE.	ND	CR10 and 1 Hz short-range telemetry system (Team 2, Polar Electro Oy, Kempele, Finland).	ND

ACC—acceleration; AMP—average metabolic power; AU—arbitrary unit; AvS—average speed; CR 10—Borg’s Category-Ratio; D—distance; DEC—deceleration; DHS—distance covered at high speed (≥18 km × h^−1^); DSS—distance covered at sprint speed (≥21 km × h^−1^); FEHS—frequency of efforts at high speed (≥18 km × h^−1^); FESS—frequency of efforts at sprint speed (≥18 km × h^−1^); GPS—global positioning systems; HR—heart rate; HR_max_—maximum heart rate; LTzone—lactate threshold; LPM—local position measurement; M—mesocycle; MEMS—micro-electrical mechanical system; MP—match-play; ND—not described; PL—player load; Pmet—equivalent metabolic power; Pmet—equivalent metabolic power; RPE—ratings of perceived exertion; SPR—sprinting; sRPE—sessions ratings of perceived exertion; sRPEmusc-TL—sessions ratings of muscular training load; sRPEres-TL—sessions ratings of respiratory training load; TD—total distance; THIA—total high-intensity activity; TL—training load; TRIMP_MOD_—modified training impulse; TS—training session; Wk—week; WRR—work:rest ratio.

**Table 5 ijerph-18-03906-t005:** Studies with predominantly focus on weekly training load distribution analysis.

Reference (Year)	Study Purpose	Periodization Structure	Independent Variable	Main Findings	Practical Applications
Abade et al. [47]	Described time–motion and physiological profile of regular training sessions.	ND	Age of players	**Distance and speed:** TD were higher in U17 (F = 45.84, *p* < 0.001). High- and very-high intensity activity were less in U19 (F = 11.8, *p* > 0.001). The number of sprints performed were different between U17 and U19 (F = −7.2, *p* < 0.001) **Accelerometry:** Total and relative body impacts were lower in U15 (F = 7.3, *p* < 0.01). **HR:** HR values showed significant effects of zone (F = 575.7, *p* < 0.001) and interaction with age group (F = −7.2, *p* < 0.001).	High variability between elite team TSs. Constrained SSG to develop basic tactical principles and technical skill may promote low physio local demands.
Akenhead et al. [53]	Described the distribution of external load during in-season 1-game weeks in in-season. Examined inter-day and interposition variation within microcycle (focus on acceleration).	Weekly microcycle (1-game week) with “match day minus” format: MD-5, MD-4, MD-3, MD-2, MD-2, MD-1, MD.	Training day and playing position	**Distance and speed:** Highest total weekly load (%) occurred on MD-4, with the lowest values on MD-1. CM covered ∼8–16% greater TD than other playing positions (excluding WM) and covered ∼17% greater distance accelerating than CD (*p =* 0.03, d = 0.7). There are associations between AvS (m × s^−2^) and the rate of accumulation for HSR, SPD, >1_TOTAL_, and >3_TOTAL_. **Acceleration:** ACC/DEC did not differ across days with the greatest variation tending to be in MD-1. No interaction between day and playing positional were found.	Monitoring only speed-based locomotor variables may not provide sufficient information about training demands. Quantification acceleration variables may add additional information.
Brito et al. [69]	Analyzed the influence of match-related contextual variables on TL and fatigue. Concomitantly, investigated if there were variations throughout the season.	Four different season phases: preparation I (3 weeks), competition I (18 weeks), preparation II (8 weeks, winter break) and competition II (12 weeks).	Contextual variables (e.g., result of previous MP, MP location, and quality of opposition).	**Distance and speed:** Weekly TLs were higher after playing a defeat or draw (*p* ≤ 0.05; d = 0.30–0.45) and after an away MP (*p* ≤ 0.05; d = 0.23). Weekly TL decreased as the season progressed (*p* < 0.001). **Perceived Exertion:** Internal load variation ranged 5 to 72% throughout the season (29–49% to weekly TL; 18–44% to fatigue scores).	Internal load variability within a season may need a more individualized approach to prepare initial and subsequent match conditions. Adding that variability together relatively stable fatigue scores may modulate pace during training.
Clemente et al. [89]	Analyzed intra-week variations during a typical weekly external load and compared variance in four professional teams.	Weekly microcycle (1-game week) with “match day minus” format: MD+1, MD + 2, MD-5, MD-4, MD-3, MD-2, MD-2, MD-1, MD.	Training day	**Distance and speed:** MD-1 had significantly less training while other days were more intense (*p =* 0.001). Portuguese team showing significantly higher intensity (SPR distance) and volume (total distance) in all days with exception of MD-1 than the Dutch team (*p* < 0.05). **Accelerometry:** Dutch team had significantly greater value of PL in MD-3 (*p =* 0.005; d = 1.18) and Portuguese team had higher PL in the MD+2 (*p =* 0.005; d = 1.78).	The training TL and tapering strategies were different between teams in different countries. However, both teams applied a significant tapering phase in the last two days before the competition in an attempt to reduce residual fatigue accumulation.
Clemente et al. [88]	Quantified weekly external load and intra-week variations during a pre-season training and compared variance in two professional teams.	Weekly microcycle (1-game week) with “match day minus” format: MD-5, MD-4, MD-3, MD-2, MD-2, MD-1, MD.	Training day	**Distance and speed:** Weekly TL presented significant differences between TS considering the duration (*p =* 0.011), walking distance (*p =* 0.017), running distance (*p =* 0.004), and number of sprints (*p =* 0.006). Variations between weeks were small and intra-week variations in the measures associated with great volume and lower intensity. **Accelerometry:** Weekly TL also presented significant differences between TS considering PL (*p =* 0.040).	Intra-week TL is not linear or standardized during in-season competition and monitoring weekly variance for the same type of day provided a useful strategy to control training adaptations.
Coutinho et al. [47]	Described the time–motion and physiological performance profiles during a typical weekly microcycle.	Weekly microcycle (1-game week) divided into: post-match (session after the match), pre-match (session before the match), and middle week (average of remaining sessions).	Age of players and weekly microcycle division (pre-match, mid-week, and post-match).	**Distance and Speed:** *U15* Mid-week showed a higher number of sprints, distance covered in intermediate speed zones, and time spent above 90% HR_max_. Pre-match presented a higher distance covered above 18 km × h^−1^ and time spent below 75% HR_max_. *U17* Pre-match and post-match presented lower distance covered values than mid-week. *U19* Post-match showed higher distance covered above 13 km × h^−1^, body impacts (>10 g), and time spent above 85% HR_max_. **Accelerometry**: *U15* body impacts showed significant differences across all TSs. *U17* pre-match and post-match presented moderate differences in body impacts. *U19* middle-week showed higher values in body impacts and pre-match presented less values than the middle-week (35% to 100%).	Appropriate physical and physiological load during middle-week TSs should be ensured. Understanding the weekly training and match load variations can contribute to optimizing short- and mid-term planning during different developmental stages.
Jeong et al. [55]	Quantified and compared TL during a preseason and in-season training process.	Season phases divided into preseason and in-season. Training mode subdivided into physical training, technical/tactical training, and physical and technical/tactical training.	Training mode/type or sub-components and season phase.	**HR and Perceived Exertion:** Preseason load was higher than in-season load (*p* < 0.05). Time spent in 80–100% maximum heart rate zones greater proportion in preseason and in-season, while technical/tactical sessions had higher intensities in the pre-season (*p* < 0.05).	Preseason is more intense than in-season training. Emphasis on higher intensities and time spent in technical/tactical specific TSs may provide the necessary physiological conditioning.
Malone et al. [56]	Quantified the seasonal TL, including both the preseason and in-season phase.	Season phases divided into preseason and in-season. Mesocycle ranged from 1 to 6 weeks (week blocks) and weekly microcycle (1-game week) with “match day minus” format: MD-5, MD-4, MD-3, MD-2, MD-2, MD-1, MD.	Season phase, mesocycle, training day and playing position.	**HR and Perceived Exertion:** typical daily TL did not differ during each week of the preseason. Daily TD covered was greater in the 1st mesocycle than in the 6th. %HR_max_ values were also greater in the 3rd mesocycle than in the 1st. TL was lower on MD-1 (regardless mesocycle) and no differences were found in other days (MD-2 to MD-5). Positional differences were found during both preseason and in-season phases. In total, CM and WD covered the highest TD. Defenders (CD and WD) displayed higher %HR_max_ values.	Quantify TL using different measures can provide physiological patterns across a full competitive season. First and last TSs optimized recovery and prevent fatigue accumulation. Positional differences should also be considered in the loading analysis.
Oliveira et al. [72]	Quantified TL using s-RPE and HI across mesocycles during an in-season comparing player positions.	Mesocycle (one month) and weekly microcycle (1-game week) with “Match day minus” format: MD-5, MD-4, MD-3, MD-2, MD-2, MD-1, MD.	Mesocycle, training day, and playing position.	**Distance and speed:** Daily TD covered was higher at the start (M1 and M3) compared to the final mesocycle (M10) of season. HSR distance was greater in M1 than M5. CM covered more distance and WM cover more distance at HSR. **Acceleration and accelerometry**: All TL variables expressed significant lower values to other days prior to a MP and no difference between player positions (*p* < 0.01). **Perceived Exertion:** Perceptual response was higher in M1 in comparison to the last mesocycle. sRPE presented a non-perfect pattern by decreasing values until MD-1: MD-5 < MD-4 < MD-3 > MD-2 > MD-1. HI showed minor variations across mesocycles and in days before MP.	Combination of different TL measures could provide evidence to fully evaluate the patterns observe across the in-season. MD-1 presented a reduction of external load (regardless of mesocycle) and HI did not change, except for MD+1.
Owen et al. [108]	Analyzed a training mesocycle whilst quantifying TL across playing position and examined the effect of match location, match status, and age of players.	Mesocycle (6 × 1-week block) and weekly microcycle (1-game week) with “match day minus” format: MD-4, MD-3, MD-2, MD-2, MD-1, MD.	Mesocycle, training day, contextual variables (match location and match status), age of players, and playing position.	**Distance and speed:** Typical daily TL did not differ throughout each week of the mesocycle in-season period. TL were significantly lower on MD-1 (*p* < 0.05). Lower AvSs were reported in training post-successful MP compared to defeats (*p* < 0.05), and more specifically when a MP was played away compared to home fixtures (*p* < 0.05). **Acceleration and accelerometry:** Significant differences in physical outputs were also found between MD-2, MD-3 and MD-4 (*p* < 0.05).	Analysis of training mesocycle and microcycle positional demands may provide useful information to training program design and tactical strategy. Physical outputs on MD-2, MD-3, and MD-4 highlighting a structured periodized tapered approach.
Rago et al. [61]	Quantified the weekly TL according to different match-related contextual factors.	Training structure included speed endurance training (e.g., repeated sprint activity) and aerobic high-intensity training (e.g., interval training). The remaining TS mainly concerned ball-possession games and team/opponent tactics. Individual/reconditioning sessions were excluded from the analysis. The periodization structure has not been described.	Contextual variables (opponent standard, match location, and match outcome).	**Distance and speed:** TD covered and HSR during training were higher in the week after playing against a bottom-level or top-level opponent compared to a medium-level opponent (*p* < 0.05). TD covered and HSR was higher in the week following a draw or a win, and higher before a loss compared to a draw (*p* < 0.05). **Acceleration:** The decrease in training volume (e.g., TD) and mechanical work (accelerations and decelerations) performed throughout the season may have been related to changes in training activities prescribed by the technical staff as a consequence of cumulative seasonal TL (*p* < 0.05).	Weekly TL seems to be slightly affected by match-related contextual variables, with special emphasis on the opponent standard and match outcome. Higher training volume was observed before and after playing against a top-level opponent, and after losing a match, whereas the volume of high-intensity training seems to be higher when preparing for a game against a top-level opponent.

>1_Total_—acceleration or deceleration ≥ 1 m × s^−2^; >3_Total_—acceleration or deceleration ≥ 3 m × s^−2^; ACC—acceleration; AvS—average speed; CD—central defenders; CM—central midfielders; DEC—deceleration; g—G force; HR—heart rate; HRmax—maximum heart rate; HSR—high speed running; M—mesocycle; MD—match day; MP—match play; ND—not described; SPR—sprinting; SSG—small-sided games; TD—total distance; TL—training load; TS—training session; TSs—training sessions; U15—under-15; U17—under-17; U19—under-19; WD—wide defenders; WM—wide midfielders.

**Table 6 ijerph-18-03906-t006:** Studies with predominant focus on weekly training load and match load distribution analysis.

Reference (Year)	Study Purpose	Periodization Structure	Independent Variable	Main Findings	Practical Applications
Anderson et al. [87]	Quantified training load during a one-, two-, and three-game week schedule.	Three different weeks: one-, two- and three-game week schedule. 1-game week: 2 days of and 4 training days before MP; 2-game week: 1 day off after 1st MP and 4 training days before second MP; 3-game week: 1 day off and training day before 1st match and the same schedule within 2nd and 3rd MP.	Weekly microcycle type	**Distance and speed:** *1-game week* TL was progressively decreased in 3 days prior to MD (*p* < 0.05). Daily TL and periodization were similar in the one- and two-game weeks. *2-game week* total accumulative distance (inclusive of both MP and TL) was higher than 1-game week. *3-game week* daily training TD was lower compared to 1-and 2-game weeks, though accumulative weekly distance was highest in this week and more time was spent in speed zones > 14.4 km × h^−1^ (*p* < 0.05).	Quantify daily training and accumulative weekly load (match load includeed) can be a support CHO periodization. Muscle glycogen is the predominant energy source and high levels of muscle glycogen may attenuate training adaptations.
Anderson et al. [106]	Quantified training load and match load during a season within starting status (starters, non-starters, and fringe).	Mesocycle (5 different in-season periods): 4 × 8-weeks (periods 1–4) and 1 × 7-weeks (period 5).	Player’s starting status (starters, non-starters, or fringe)	**Distance and speed:** Starters completed more moderate intensity running, HSR, and SPR than non-states (*p* < 0.01). Starters also completed more SPR than fringe players (*p* < 0.01).	Seasonal volume and intensity training are dependent on player’s match starting status and must be considered for training program design.
Baptista et al. [97]	Quantified the most demanding passages of play in training sessions and matches (5-min peaks); and evaluated the accumulated load of typical microcycles and official matches, according to playing position.	Weekly microcycle (1-game week) with “match day minus” format: MD+1C, MD+1R, MD-4, MD-3, MD-2, MD-2, MD-1, MD	Playing position and weekly microcycle.	**Distance and speed:** Training values for SPR and HSR distance were lower (36–61% and 57–71%) than for acceleration variables. The highest difference was verified on the 5-min peaks for SPR_peak_, with FB achieving during the microcycle only 64%, while CB, CM, and FW levelled and overperformed the match values (107%, 100%, and 107%, respectively). **Acceleration:** Correlations match demands were overperformed for ACC counts (131–166%) and DEC counts (108–134%), according all position. Training values were higher than SPR and HSR distance.	Differences observed across playing positions in matches and microcycles underline the lack of position specificity of common training drills/sessions. Coaches and practitioners must keep in mind that the absolute TL accumulated by players of different positions, so analyzing the relative TL (according to the match demands) may be a much better and more valuable way of managing and evaluating the players periodization.
Dalen et Lorås [102]	Analyzed physical (locomotor activities) and physiological (Banister’s training impulse) in-season training load between starters and substitutes.	ND	Player’s starting status (starters and non-starters)	**Distance and speed:** Starting players demonstrated significantly higher average weekly physical load compared to the non-starters for all variables: TD, running, HSR, and SPR (*p* < 0.001), number of ACC and SPR (*p* < 0.001). Similarly, Banister’s TL (*p* < 0.001) was significantly higher within week than starters than non-starters.	The weekly accumulated high-speed running and sprint distances were largely related to match playing time. Therefore, weekly fitness-related adaptations in running at high speeds seem to favor the starters in a soccer team.
Clemente et al. [95]	Described the training/match ratios and variations between different weekly microcycle type. Investigated relationship within weekly accumulated TL and match load.	Three different weekly microcycle: week with 5 TSs (5 dW), 4 TSs (4 dW) or 3 TSs (3 dW).	Weekly microcycle type	**Distance and speed:** TDr, HSRr, and SPRr were significantly greater in 5 dW (*p* < 0.001). **Acceleration and accelerometry:** Correlations between the weekly TL and the match load of the same week were small for PL (r = 0.25 (0.13; 0.36)), ACC (r = 0.29 (0.17; 0.40)) and DEC (r = 0.23 (0.11; 0.35)).	Additional TSs, it may be necessary to promote differences between weekly accumulated TL and the load imposed in a single MP. Relationship between weekly accumulated TL and weekly MP are dynamic and unpredictable which may be impossible for accumulated weekly TL and their variations to be adjusted according to match loads.
Clemente et al. [105]	Analyzed the variations of acute load, training monotony, and training strain among pre-season, mid-season and end-season according playing position.	Mesocycle (5 different in-season periods): (i) pre-season (week 1 to week 6); mid-season or first half of the season (week 6 to week 33); and end-season or second half of the season (week 34 to week 45).	Season phase	**Distance and speed:** Training monotony and training strain for HSR were meaningfully greater in pre-season than in the mid-season and greater than the end-season (*p* < 0.001). The training monotony for the sprints was meaningfully greater in pre-season than in the mid-season and greater than the end-season (*p* < 0.001). Comparisons between playing positions revealed small-to-moderate effect size differences mainly for the number of sprints in acute load, training monotony, and training strain.	Acute load, training monotony, and training strain occurred in the pre-season and progressively decreased across the season. Moreover, external defenders and wingers were subjected to meaningfully greater acute load and training strain for HSR and number sprints during the season compared to the remaining positions.
Kelly et al. [70]	Analyzed TL and match load across a full season.	Mesocycle ranged from 6 to 9 weeks.	Mesocycle and playing position	**Distance and speed:** Daily TDs were higher during the early stages (M1 and M2) of the competition period. Overall, high-speed activity was similar between mesocycles. Weekly TL was greater on MD and lower MD-1 (*p* < 0.001). TD progressively decreased over the MD-3 (*p* < 0.001). High-speed distance was greater MD-3 while very high-speed distance was greater on MD-3 and MD-2 than MD-1 (*p* < 0.001). **Perceived Exertion:** Daily sRPE was also higher across early season stages. sRPE progressively decreased over the MD-3 (*p* < 0.001) as well as TD (*p* < 0.001).	Limited TL variation across mesocycles suggests that training schedules employed a highly repetitive likely reflecting the nature of the competition demands. TL periodization included a three-day period leading into competition.
Los Arcos et al. [71]	Quantified and compared the respiratory and muscular perceived TL accumulation depending on the player participation.	Mesocycle ranged from 6 to 8 weeks (week blocks) and weekly microcycle (1-game week) with “match day minus” format: MD-5, MD-4, MD-3, MD-2, MD-2, MD-1, MD.	Player’s starting status (starters or non-starters), mesocycle and training day.	**Perceived Exertion:** Weekly TL variation across the in-season blocks was trivial-small (except between block 2-block3). Substantial TL differences were found between training days. Weekly TL was a progressive increase up to MD-3 followed by a decrease until MD-1. sRPEres/sRPEmus-TL reported during MD was very similar between starters and non-starters.	Perceived TL across the season displayed limited variation. Highest weekly TL was applied to 72 h before the MD to progressively between MD-3 and MD.
Martin-Garcia et al. [82]	Determined the external load across playing position and relative for a structured microcycle. Examined TL and variation the day after competition for players with or without MP time.	Weekly microcycle (1-game week) with “match day minus” format: MD+1C, MD+1R, MD-4, MD-3, MD-2, MD-2, MD-1, MD.	Playing position and training day.	**Distance and speed:** TL declined as competition approached (MD-4 > MD-3 > MD-2 > MD-1; *p* < 0.05). MD+1C by players with game time was higher than MD+1R (*p* < 0.05). FB performed more high-speed running and SPR than other positions at MD-3 and MD-4 (*p* < 0.05; (0.8–1.7)). Weekly TL variation was ~40% for MD-3 and MD-4 to ~80% for MD+1R. **Acceleration:** ACC/DEC represented 50% of that performed in competition for MD+1C (80–86%), MD-4 (71–72%), MD-3 (62–69%), and MD-2 (56–61%). **Metabolic power:** MD+1C demonstrated greater HLMD and AMP than MD+1R (*p* < 0.05; (1.4–1.6)). TL declined as competition approached (MD-4 to MD-1) as well as HMLD and AMP (*p* < 0.05; ES: 1.5–3.0).	Quantifying TL should consider the relative competition demands and position-specific loads. MD+1 can be used to compensate for the reduced competition load in player with limited playing time. MD-4 and MD-3 could be employed to elevated training stimulus.
Owen et al. [99]	Investigated multi-metric monitoring method highlighting TL and its relationship to MP.	Weekly microcycle (1-game week) with “match day minus” format: MD-5, MD-4, MD-3, MD-2, MD-2, MD-1, MD.	Training day	**Distance and speed:** Significant differences between daily TL and competitive TL. Additionally, significant differences between training days for both volume- and intensity session scores (*p* < 0.05). No differences were found between MD-1 and MD-2 session scores.	Specific multi-modal approach may combine key mechanical volume and intensity metrics to player monitoring strategies and tapering approaches. The TL and match load relationships could provide a better understanding to the need for prepare players individually in line with MP demands.
Sanchez-Sanchez et al. [91]	Quantified the external load during in-season training microcycles and examined its relationship to the competition demands.	Weekly microcycle (1-game week) with “match day minus” format: MD-5, MD-4, MD-3, MD-2, MD-2, MD-1, MD.	Training day	**Distance and speed:** External load variables were similar between the four microcycles. MD-2 presented highest TL on time, TD, HSR and SPR compared MD+1, MD-3 and MD-1 (*p* < 0.01). **Acceleration:** Aside from training duration, all external loads variables were lower during training sessions compared to official matches (*p* < 0.05).	Absolute and relative external load values allow to more accurately know the load applied. MP constitutes the highest load during a typical competitive microcycle and MD-2 contain the weekly peak load.
Swallow et al. [110]	Quantified the external TL across both training and competitive matches during the season. Examined the influence of one and two match weekly microcycles on the external TL.	Weekly microcycle (1-game week) with “match day minus” format: MD-5, MD-4, MD-3, MD-2, MD-2, MD-1, MD.		**Distance and speed:** TD and HSR were higher on MD and MD-5. MD-4 displayed significantly higher values compared to MD-1 and MD-2. The 2-game week presented a TD higher on MD-1 when compared to 1-game week. However, lower values were observed for duration and HSR on MD-2 and MD-4 during the 2-game week compared to the 1-game week. **Acceleration:** Higher values recorded on MD for number of ACC. ACC data were influenced for the different game week schedule. **Accelerometry:** PL was also higher on MD and MD-5. The 2-game week presented a higher PL on MD-1 than 1-game week.	Progressive reduction in TD, PL, HSR, and ACC leading into competitive matches based on MD- analysis. However, some variability exists in TL prescription as a result of different 1-game week schedules (i.e., 1-game week vs. 2-game week).
Stevens et al. [80]	Quantified and compared the TL of training days and MP. Compared training of nonstarters the day after the match with regular training of starters and non-starters.	Weekly microcycle (1-game week) with “match day minus” format: MD-5, MD-4, MD-3, MD-2, MD-2, MD-1, MD.	Player’s starting status (starters or non-starters)	**Distance and speed:** TL was lower when training approached MD. Training values for running and HSR were lower than for TD, and all considerably lower than MD values. Non-starters training was lower loading than regular training for almost all variables on MD-4 and several high-intensity variables on MD-3 and MD. **Acceleration and metabolic power:** Medium and high accelerations and decelerations during training were more similar to match values. MD-4 was the greatest TL, including acceleration and metabolic variables.	Acceleration load on the most intense training day in MD-4. Non-starters training showed in a more general load than regular training, especially on MD-4, contributing to a considerably lower total weekly TL for non-starters. There is a challenge to improve sufficiently load in non-starters, especially in terms of running and HSR.
Wrigley et al. [60]	Quantified typical weekly TL during the in-season competitive period.	Weekly microcycle (1-game week): Monday, Tuesday, Wednesday, Thursday, and Saturday or Sunday (MD).	Age of players, training day and training mode/type or sub-components	**HR and Perceived Exertion:** Total weekly TL (training and match) increased with (*p* < 0.05). Differences in the daily TL across the training week were also evident in the older age group (U18). The amount of time engaged in low (<50% HRmax) and high (>90% HR_max_) intensity activity during training and match-play was significantly lower and higher respectively in the U18 compared to the U14 group (*p* < 0.05). When comparing activity, the intensity (% HR_max_) of field training was significantly lower compared to MP across all age groups (*p* < 0.05).	Age-related increases reflect increases in the intensity and a greater extent of the training volume. Weekly periodization in an older player may adopt an exponential tapering focused on competition.

1st—first; 2nd—second; 3 dW—week with three training sessions; 3rd—thirty; 4 dW—week with four training sessions; 5 dW—week with five training sessions; ACC—acceleration; AMP—average metabolic power; CB-centre back; CHO—carbohydrate; CM-center midfielders; DEC—deceleration; FB—full-backs; FW—forwards; g—G force; HMLD—high metabolic load distance; HR_max_—maximum heart rate; HSR—high speed running; HSRr—high speed running ratio; M—mesocycle; MD—match day; MD+1C—match day + 1 compensatory; MD+1R—match day + 1 recovery; MP—match play; PL—player load; SPR—sprinting; SPR_peak_—sprint peak; SPRr—sprinting ratio; sRPE—sessions ratings of perceived exertion; sRPEmusc-TL—sessions ratings of muscular training load; sRPEres-TL—sessions ratings of respiratory training load; TD—total distance; TDr—total distance ratio; TL—training load; TS—training session; TSs—training sessions; U14—under-14; U18—under-18.

**Table 7 ijerph-18-03906-t007:** Studies with predominant focus on the relationships between internal and external load.

Reference (Year)	Study Purpose	Periodization Structure	Independent Variable	Main Findings	Practical Applications
Alexiou and Coutts [52]	Compared the sRPE method for quantifying internal load with various HR-based TL quantification (Bannister’s TRIMP, LTzone TL and Edward’s TL) in different training modes.	Weekly microcycle (1-game week): 3 TSs technical/tactical, 2 TSs high-intensity resistance, 1 TS aerobic conditioning, 1 TS core stability, 1TS pool “recovery” and 1 MP.	Training mode/type or sub-components	**RPE vs. HR:** Correlation for RPE and method for quantifying internal load were: Bannister’s TRIMP (r = 0.84); LT zone (r = 0.83); Edwards TL (r = 0.85). There are differences between training mode (*p* < 0.001); strongest correlations were reported for technical (r = 0.68 to 0.82), conditioning (r = 0.60 to 0.70) and speed (r = 0.61 to 0.79) sessions.	sRPE method was a valuable tool to internal load quantification that can measure both psychological and physiological factors. Therefore, sRPE seems to be a more global indication of the internal stress.
Campos-Vazquez et al. [50]	Described internal load performed during a typical week and determined the relationship between different internal load measures.	ND	Training mode/type or sub-components	**RPE vs. HR:** Higher internal load during skills drills/circuit training and small sided games than in tactical training and pre-match activation. Large relationships were found between HR_max_ > 80% HR_max_ and R > 90% HR_max_ vs. sRPE (r = 0.61 to 0.68). Very large relationships were found between Edwards TL and sRPE and between TRIMP_MOD_ and sRPE (r = 0.73 to 0.87).	Internal load variables relationships differ according training mode/type. For this reason, caution should be applied when using RPE- or HR-derived measures to quantify training or exercise intensity.
Casamichana et al. [115]	Examined the relationship between internal and external load indicators used to quantify TL.	ND	TL indicators (external and internal load)	**PL vs. HR and RPE:** Very-large association for PL with Edward’s TL (r = 0.72, *p* < 0.01) and sRPE (r = 0.76, *p* < 0.01). TD vs. PL and RPE: Large to very-large association between TD and PL (r = 0.70, *p* < 0.01), sRPE (r = 0.74, *p* < 0.01) and (r = 0.72, *p* < 0.01).	sRPE was a global indicator to measure internal training response. Very large association between PL and internal load measures expresses the interest of accelerations monitoring. TL analysts should take advantage using GPS technology and sRPE or Edwards methods for post-hoc TL monitoring.
Gaudino et al. [90]	Compared measurements of high-intensity activity during field-based TS in different playing positions. TD covered at >14.4 km × h^−1^) and TP (>20 W × kg^−1^).	ND	Playing position	**TD vs. TP:** Difference within TD covered at >14.4 km × h^−1^ and TP was greater for central defenders (~85%) than WD and attackers (~60%, *p* < 0.05). Differential between methods also decreased as the proportion of high-intensity distance within a training session increased (R^2^ = 0.43, *p* < 0.001).	Metabolic power may provide better examination for high-intensity component of training which typically represents the most physically demanding elements. Including metabolic power analysis can minimize underestimation on external load quantification using traditional monitoring approach.
Gaudino et al. [118]	Identified the external load measures that are most influential on perceptual response during training sessions.	ND	TL indicators (RPE)	**RPE vs. HSR:** Perceptual responses provided within-individual correlations with HSR (*p* < 0.001). **RPE vs. body impacts:** RPE within correlated with the number of impacts (*p* < 0.001). **RPE vs. ACC:** Within-individual correlations with ACC (*p* < 0.001).	HSR, the number of impacts and accelerations are the best external load measures to predict perceptual response during training process. Understanding the influence of characteristics affecting RPE may help in enhance training design and athlete monitoring.
Impellizzeri et al. [54]	Quantified internal load using sRPE and assessed correlations within HR-based methods (Edwards, Banister, and Lucia TL).	Weekly microcycle (1-game week): Monday, Tuesday, Wednesday, Thursday, and Saturday (MP). Sunday and Friday are days ff. Typical training program was: heaviest aerobic training (Monday), speed developing through sprint and plyometric (Tuesday), running interval training (2 times week), and MP (Saturday).	Training day	**RPE vs. HR**: Mean sRPE reported to field-based training was: Monday (32%), Tuesday (27.8%), Wednesday (22.8%), and Thursday (17.3%). Match load corresponded to 24% of the total weekly TL. Peak internal load was reached the first day of the training week (after a day of total recovery). Individual TS showed some variability on peak internal TL sessions within the week. All individual correlations between various HR-based TL and sRPE were statistically significant (r = 0.50 to 0.85, *p* < 0.01).	sRPE can be considered a good indicator to global internal load and has potential to TL quantification. The moderate correlation cannot support this method as a HR-based methods substitute, as only about 50% of variance in HR was explained by sRPE.
Kelly et al. [65]	Quantified the within-participant correlations between variability in sRPE and HR-derived measures.	ND	Playing positions	**RPE vs. HR:** The correlation between changes in sRPE and Edwards TL (r = 0.75). These correlations across playing position: WD (r = 0.81); CD (r = 0.74); WD (r = 0.70); CM (r = 0.70); ST (r = 0.84) (*p* < 0.001).	sRPE was a simple and practical global indicator of individual TL in elite-level soccer player regardless the playing position.
Marynowicz et al. [98]	Examined the relationship between the external TL markers and the RPE and session-RPE (sRPE), thereby identifying those that are most influential.	ND	TL indicators (external and internal load)	**RPE vs. sRPE:** Large, positive within-individual correlations (r = 0.62, *p* < 0.001). **RPE vs. TD:** Large, positive within-individual correlations (r = 0.70, *p* < 0.001). **RPE vs. HSR:** Moderate within-individual correlation (r = 0.39, *p* < 0.001). **RPE vs. ACC:** Large, positive within-individual correlations (r = 0.64, *p* < 0.001). **RPE vs. PL:** Large, positive within-individual correlations (r = 0.70, *p* < 0.001).	The findings demonstrate that RPE does not reflect the intensity of a training session and that sRPE can be a useful, simple, and cost-effective tool for monitoring TL. Determining which external load markers have the most influence on the perception of effort enables coaches to better monitor athletes and as a consequence both reduce the risk of injury and improve physical performance.
Rago et al. [107]	Examined the within-player correlation between perceptual responses (RPE) and external load (high-speed running using arbitrary and individualized speed zones).	Weekly microcycle (1-game week): “match day minus” format: MD-5, MD-4, MD-3, MD-2, MD-2, MD-1, MD. Day after MP was day-off.	Training day	**RPE vs. HSR and SPR:** Moderate correlation for perceptual responses within MSR and HSR quantified using the arbitrary method (r = 0.53 to 0.59; *p* < 0.05). The magnitude of correlations tended to increase when the individualized method was used (r = 0.58 to 0.67; *p* < 0.05). Correlation to SPR was moderate only when the individualized method was used (0.55 (0.05; 0.83) and 0.53 (0.02; 0.82), *p* < 0.05). **RPE vs. HSR and SPR:** Perceptual responses were largely correlated to TD within all three speed running zones, independently quantification method (r = 0.58 to 0.68; *p* < 0.05). No significant correlations were observed when external load was measured with percentage (*p* > 0.05).	Adjusted values of distances covered within the TSs for individual speed being more representative of perceptual responses to training, rather than percentage of TD. Instead, splitting values of distances covered can provide better information about individual perceptual responses to the training process.
Scott et al. [57]	Compared various measures of training load derived from physiological and physical data during in-season field-based training.	ND	TL indicators (external and internal load)	**HR and RPE vs. TD, PL and HSR/SPR:** Large correlation for TD, LSA volume, and PL with HR-based and sRPE-based methods (r = 0.71 to 0.84; *p* < 0.01) correlations. Moderate to large correlation for HSR volume and very HSR with measures of internal load (r = 0.40 to 0.67; *p* < 0.01).	TD, LSA, and PL can be useful external load indicators to field-based training. Physical activity measures such HSR and very HSR may provide additional information not reflected in perceptual and physiological methods.
Vahia et al. [57]	Analyzed the in-season variation in correlation between HR-based method and perceptual response (sRPE).	Weekly microcycle (1-game week): Monday, Tuesday, Wednesday, Thursday, and Saturday (MP). Sunday is a day off. Typical training program was: 2 technical sessions (Monday), strength training (Tuesday), resistance training (Wednesday), 1 speed and technical session (Thursday), match preparation (Friday), and MP (Saturday).	Months of the season (mesocycle)	**RPE vs. HR:** The monthly correlations ranged from r = 0.60 to 0.73 (*p* < 0.05) and the overall correlation was r = 0.64 (0.60–0.68; *p* < 0.001). The changes in HRTL and sRPE showed large correlations over months (r = 0.64 [0.60–0.68]; *p* < 0.001)	sRPE was a reliable measure to measure internal load during the entire season. This method presented small variations and little bias when compared to HR-derived methods.

ACC—acceleration; CD—central defenders; CM—central midfielders; HR—heart rate; HRmax—maximum heart rate; HRTL—heart rate training load; HSR—high speed running; LSA—low speed activity; LT—lactate threshold; MP—match play; MSR—moderate speed running; ND—not described; PL—player load; RPE—rating of perceived exertion; SPR—sprinting; sRPE—session rating of perceived exertion; ST—strikers; TD—total distance; TP—equivalent metabolic power of >20 W × kg^−1^; TL—training load; TRIMP—training impulse; TRIMP_MOD_—modified training impulse; TS—training session; TSs—training sessions; WD—wide defenders; WM—wide midfielders.

## Data Availability

Data is available under request to the contact author.

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
