# Peer review of "Monitoring Accumulated Training and Match Load in Football: A Systematic Review"

_ijerph, 2021, doi:10.3390/ijerph18083906_

Round 1

Reviewer 1 Report

I really appreciate the opportunity to review this manuscript entitled “Monitoring Accumulated Training and Match Load in Football: 2 A Systematic Review”.  I only remark some issues (most of them in methods) in order to improve the quality of this manuscript.

The abstract is clear but it is important to explain in the main text (mainly in the discussion) why the study is mainly about men.

At the methods section, there are some questions that should be review. Related to the search strategy, is there any reason about not choosing Sportdiscus database for the searh? About the inclusion criteria, the authors take into account that participants are older than 14 but, should they be professionals? Regarding the exclusion criteria, number 7 is about articles with bad quality according to STROBE, which score is consider bad quality?

Results were clear. Conclusions and references were correct.

Author Response

Thank you very much for the time you spent and your feedback on this manuscript. We have made every effort to take on board your recommendations and comments. We hope this 2nd revised version and the responses to the comments (kindly refer to our replies below) will meet your requirements. Please note that all new changes in the revised manuscript are edited with the Microsoft Word® tracking tool.

Reviewer 2 Report

First of all, I want to thank the authors for the effort to develop this systematic review. I will pinpoint some aspects after my review:

  • Abstract: the final number of articles considered for the systematic review is different in the abstract, the flowchart and the manuscript (line 229)
  •  Abstract: the access to the literature search shows inconsistency between abstract, manuscript and the ine 132
  • Manuscript: a moderate review of the english language should be carried out. Being a non native english reviewer, I have been able to follow the manuscript but I have detected some areas needing a review or that I have not fully understood (e.g., line 45 "fitness, which..."; Line 202 "the references it was provided"; line 242 "geograph"; line 402 "The lowest load it was found..."; line 507 "e"; line 554 "it´s possible ensure"; line 559 "Evaluate"; line 587 "According these"; line 630 "Determine..."; line 660 "....has"; ) or that I have not fully understood (lines 391-394; lines 483-485; line 665 "(9   n    160)"; line 669 "since to these date")
  • The use of acronyms in the discussion should be at least explained the first time they appear. E.g., I have not found explanation for AU (line 549)
  • There is a inconsistency of using present and past throughout the text.
  • Table 1 title and explanation are not appropiate ("This is a table. Tables should be placed in the main text near to the first time they are cited)
  • Limitations. Only english language has been used for the systematic review. A language limitation should be considered for not including other languages
  • I have felt a quite long manuscript. Making a mini-conclusion at the beginning of each section of the discussion would be thanked by the readers

Author Response

Thank you very much for the time you spent and your feedback on this manuscript. We have made every effort to take on board your recommendations and comments. We hope this revised version and the responses to the comments (kindly refer to our replies below) will meet your requirements. Please note that all new changes in the revised manuscript are edited with the Microsoft Word® tracking tool.

Round 2

Reviewer 2 Report

Please, verify the final result of the tracked changes and ortography when the document is clean.